# Beyond Rewards: a Hierarchical Perspective on Offline Multiagent Behavioral Analysis

**Shayegan Omidshafiei**
somidshafiei@google.com

**Andrei Kapishnikov**
kapishnikov@google.com

**Yannick Assogba**
yassogba@google.com

**Lucas Dixon**
ldixon@google.com

**Been Kim**
beenkim@google.com

Google Research

## Abstract

Each year, expert-level performance is attained in increasingly-complex multiagent domains, where notable examples include Go, Poker, and StarCraft II. This rapid progression is accompanied by a commensurate need to better understand how such agents attain this performance, to enable their safe deployment, identify limitations, and reveal potential means of improving them. In this paper we take a step back from performance-focused multiagent learning, and instead turn our attention towards agent behavior analysis. We introduce a model-agnostic method for discovery of behavior clusters in multiagent domains, using variational inference to learn a hierarchy of behaviors at the joint and local agent levels. Our framework makes no assumption about agents' underlying learning algorithms, does not require access to their latent states or policies, and is trained using only offline observational data. We illustrate the effectiveness of our method for enabling the coupled understanding of behaviors at the joint and local agent level, detection of behavior changepoints throughout training, discovery of core behavioral concepts, demonstrate the approach's scalability to a high-dimensional multiagent MuJoCo control domain, and also illustrate that the approach can disentangle previously-trained policies in OpenAI's hide-and-seek domain.

## 1 Introduction

Multiagent approaches have driven numerous advances in artificial intelligence research, with seminal examples including TD-gammon [1], DeepBlue [2], AlphaGo [3], AlphaZero [4], Libratus [5], AlphaStar [6], OpenAI Five [7], and Pluribus [8]. During training, many of these approaches seek to push the performance of agents as measured by a reward signal, or derivatives thereof.

Despite this, post-hoc methods that seek to *understand* agent interactions often use less reward-centric techniques. Instead, insights are drawn from behavioral analysis to identify unique or interesting agent strategies. Examples include clustering-based analysis of neuron activations and trajectories in capture-the-flag [9], inspection of trajectories in a hide-and-seek domain to detect interesting behaviors such as agents that learn to exploit the underlying physics engine [10], monitoring of statistics such as pass ranges and frequencies in humanoid football [11], and analysis of AlphaZero's acquisition of chess knowledge [12]. Crucially, such insights are often drawn via manual analysis and detection of behavioral clusters, or use of statistics associated with certain behaviors as defined by humans experts (e.g., various skills and relevant metrics in humanoid football).

36th Conference on Neural Information Processing Systems (NeurIPS 2022).

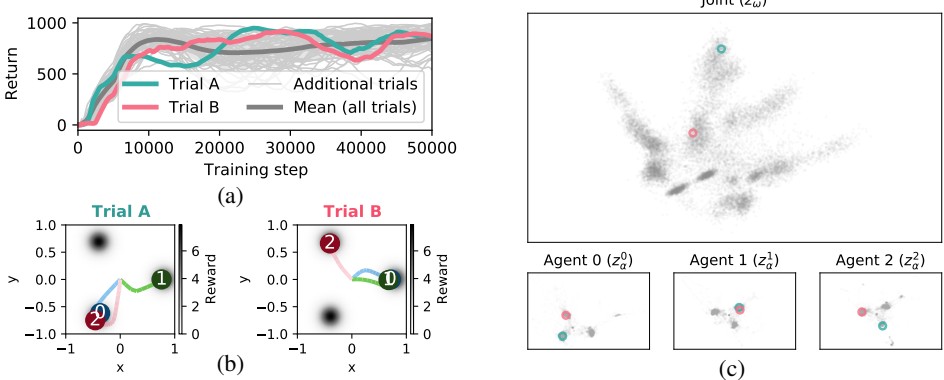

Figure 1: Reward alone is not enough to understand underlying behaviors in a 3-agent hill-climbing domain. Agents here start at the origin, each receiving rewards by navigating to any of 3 equidistant hills. (a) visualizes the total returns of agents throughout training, over 50 independent trials. Two trials (A and B) with similar final returns are highlighted. (b) visualizes the actual converged behaviors of the agents at the end of Trials A and B, which are distinct despite their similar returns. Visualizing these same trajectories in the behavior space learned by our approach immediately reveals differences in the joint behavior of agents in the top panel of (c), where the two color markers correspond to the trajectories from each trial. Simultaneously analyzing the agent-wise latent spaces in the bottom 3 panels of (c) highlight that agent 1 behaves the same way in both trials, in contrast to agents 0 and 2.

As evident above, understanding emergent multiagent behaviors is enriched by techniques beyond pure reward-based analysis, as behavioral signifiers are not always discernible via rewards. Figure 1a provides intuition on this notion, illustrating returns (sum of rewards) throughout training for a multiagent hill-climbing domain (later described in detail). We highlight two independent training trials (A and B) with similar final returns. Despite similar returns, comparing the trajectories generated via the agents' deterministic policies following each trial's training (Fig. 1b) reveals entirely different behaviors. Analogous examples are evident in the above works (e.g., Fig. 1 of Baker et al. [10], where substantial behavior changes occur in multiagent hide-and-seek despite a smooth reward curve).

This paper formalizes the problem of offline multiagent behavior analysis. Our proposed algorithm, Multiagent Offline Hierarchical Behavior Analyzer (MOHBA), learns a hierarchical latent space that simultaneously reveals behavior clusters at the joint level (i.e., interactions *between* agents) and local level (i.e., behaviors of *individual* agents). Our method is agnostic of the underlying algorithm used to generate agents' behaviors, requires no access or control of the underlying environment, does not assume availability of a reward signal, and does not require access to agents' models or internal states. Our experiments investigate the structure of the learned behavior space, which goes beyond prior works on latent-clustering by identifying relationships between individual agent and joint behaviors. We illustrate that clusters identified by MOHBA are useful for highlighting similarities and differences in behaviors throughout training. We also quantitatively analyze the completeness of discovered behavior clusters by adopting a modified version of the concept-discovery framework of Yeh et al. [13] to identify interesting behavior concepts in our multiagent setting. We then test the scalability of our approach by using it for behavioral analysis of several high-dimensional multiagent MuJoCo environments [14]. Finally, we evaluate the approach on the open-sourced OpenAI hide-and-seek policy checkpoints [10], confirming that the behavioral clusters detected by MOHBA closely match those of the human-expert annotated labels provided in their policy checkpoints.

## 2  Related Work

Significant research has been conducted in single-agent skill discovery, which seeks to learn reusable policies useful for downstream tasks [15–28]. Related approaches discover motor primitives to express longer-horizon policies [29], including use of offline reinforcement learning (RL) for learning useful behaviors [30]. Option discovery methods learn temporally-abstracted actions (i.e., options [31]), chained together to form cohesive skills [32–37]. In contrast to our work, these approaches focus on maximizing performance in single-agent settings. There also exists a related line of work for learning diverse policies in RL settings [38–46]; despite their focus on policy diversity, several of these approaches make much stronger assumptions than ours (e.g., access to the policies generating agent trajectories, or unique identifiers of the policy that generated each trajectory). By contrast, our

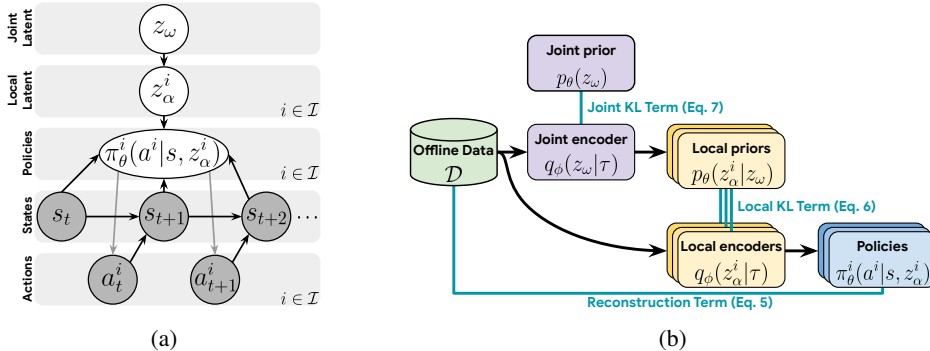

Figure 2: Approach overview. (a) Graphical model of the latent-conditioned trajectory generation process that MOHBA uses to learn multiagent behavior clusters. The joint behavior latent parameter $z_\omega$ informs local (agent-wise) behavior parameters $z_\alpha^i$, which affects their behavioral policies. Given a state-action trajectory dataset, our approach learns these joint and local behavior spaces. (b) Architecture of the MOHBA model with variational lower bound terms (5) to (7) indicated.

work assumes no access to any of the underlying raw policies, or even labels of which policies agent trajectories were obtained from. Recent works have also focused on hierarchical skill learning in multiagent reinforcement learning (MARL). Lee et al. [47] use the mutual information maximization objective introduced in Eysenbach et al. [16] to learn multiagent policies. Yang et al. [48] use a bi-level policy to learn agent skills: a high-level policy first generates latent vectors for each agent, and a low-level policy conditions behaviors on said vector to perform the task. In Wang et al. [49], distinct 'roles' are learned for agents to enable decomposition of tasks. Mao et al. [50] investigate use of consistent latent cognition variables in agent neighborhoods to induce increased cooperation. Many MARL approaches have noted the emergence of interesting behaviors in multiagent systems in specific domains of interest [51–55]. In contrast to our focus, these approaches use RL to maximize agent performance, rather than understand arbitrary behaviors via offline analysis.

RL interpretability methods primarily focus on single-agent settings and either modify the RL algorithm itself to increase transparency, or conduct post-hoc explainability [56]. These approaches represent agent policies as programming languages [57], extract visual summaries of behaviors using 'interestingness' statistics such as uncertainty in selected actions or the value of state-transitions [58], or combine agent neuron activations with gradient information to construct behavioral embeddings [59]. Behavioral clusters in our work share similarities with concept-based explanation approaches in non-RL domains. Detecting 'concepts' in pre-trained models has been explored in vision [60–62], discrete games [12], and language [63, 64]. In vision, clustering-based approaches describe discovered concepts using examples [60] or use generative modeling to create new data to describe concepts [62]. Ghorbani et al. [60] uses a vision-specific method (i.e., superpixels) to sub-divide input before conducing clustering; while sub-division is less natural in RL, our LSTM and VAE baselines serve as an RL-adopted counterpart of such works.

Works using latent-clustering and analysis are also related to ours. These include approaches using multi-level variational autoencoders [65–68] to learn compositional latent spaces, although not in decision-making domains such as ours. Hierarchical latent approaches have been used in single-agent RL [69, 70]. Behavior analysis has also been conducted by embedding agent neuron activations into a low-dimensional space, using a (non-hierarchical) variational approach [9]. The representation power of agents' internal states has also been gauged by predicting future events [11]. Overall, the key difference between the above works and ours is that our method combined hierarchical learning with behavior analysis and applies it to the multiagent setting.

## 3  Offline Analysis of Multiagent Behaviors

This section introduces the offline multiagent behavioral analysis problem and our proposed algorithm.

**Preliminaries.**  We first formalize the problem of offline multiagent behavioral analysis. Consider a rewardless multiagent Markov Decision Process (MA-MDP), defined by tuple $(\mathcal{I}, \mathcal{S}, \mathcal{A}, \mathcal{T})$, where $\mathcal{I} = \{1, ..., N\}$ is the set of $N$ agents, $\mathcal{S}$ is the state space, $\mathcal{A}$ is the action space, and $\mathcal{T}$ denotes the state transition probability function. By not relying on the presence of rewards, behaviors generated even without reliance on a reward function (e.g., human interactions, or agents using curiosity-based

exploration) can be considered. We use the term 'local' for elements associated with individual agents, and 'joint' for those associated with the entire system. At each timestep $t \in \{0, \ldots, T-1\}$, the agents execute joint action $a_t \in \mathcal{A}$ in state $s_t$ using joint policy $\pi(a_t|s_t)$, causing the state to transition to $s_{t+1}$ with probability $p(s_{t+1}|s_t, a_t) = \mathcal{T}(s_{t+1}, a_t, s_t)$. As standard in multiagent frameworks [71], we assume the joint action space factorizes as $\mathcal{A} = \times_i \mathcal{A}^i$, such that $a_t = (a_t^1, \ldots, a_t^N)$, where $i \in \mathcal{I}$ and $a_t^i \in \mathcal{A}^i$. Similarly, $\pi^i(a_t^i|s_t)$ is the local policy for agent $i$, and $\pi = \prod_i \pi^i$ is the joint policy.

Let $\tau = (s_0, (a_0^i)_{i \in \mathcal{I}}, \ldots, s_{T-1}, (a_{T-1}^i)_{i \in \mathcal{I}}, s_T)$ denote a trajectory induced by this process, and $\mathcal{D} = \{\tau_1, \ldots, \tau_K\}$ denote a dataset of $K$ such trajectories. This dataset may consist of trajectories from multiple training runs, including variations over agent algorithms, hyperparameters, random seeds, or other factors influencing emergent behaviors. Given dataset $\mathcal{D}$, the offline multiagent analysis problem seeks to uncover potential clusters of agent behaviors.

**Approach.** Our approach, called the Multiagent Offline Hierarchical Behavior Analyzer (MOHBA), uses offline trajectory data to discover behaviors exhibited by agents at the local and joint level.

We first use Fig. 2a to build intuition before discussing technical details. Let the agent interactions exhibited in a trajectory $\tau$ be encoded by a latent variable, $z_\omega \in \mathbb{R}^{D_\omega}$, capturing their joint behavior. For example, $z_\omega$ may encode (at a high level) whether agents were cooperating or competing in a given trajectory; conditioned on joint signal $z_\omega$, each agent then exhibits its own local behavior. Let local latent vectors $z_\alpha = (z_\alpha^1, \ldots, z_\alpha^N)$ encode individual agent behaviors, where $z_\alpha^i \in \mathbb{R}^{D_\alpha^i}$. Conditioned on the local behavior vector $z_\alpha^i$, each agent then executes actions using a behavior-conditioned policy $\pi^i(a^i|s, z_\alpha^i)$. Given trajectory dataset $\mathcal{D}$, we seek to learn the latent-conditioned policies and distributions over latent vectors, such that we can reconstruct *any* behaviors exhibited by the agents in $\mathcal{D}$. Thus, latent vectors $z_\omega$ and $z_\alpha$ will encode the agents' behavioral spaces and, ideally, identify behavioral clusters in the dataset. Given this framework, the joint policy is decomposed,

$$\pi(a_t|s_t) = \int_{z_\alpha, z_\omega} \pi(a_t|s_t, z_\alpha) p(z_\alpha|z_\omega) p(z_\omega) dz_\alpha dz_\omega \tag{1}$$

$$= \int_{z_\alpha, z_\omega} \prod_{i=1}^N \pi^i(a_t^i|s_t, z_\alpha^i) p(z_\alpha^i|z_\omega) p(z_\omega) dz_\alpha dz_\omega , \tag{2}$$

where in (1) we have assumed that each agent's latent-conditioned policy is conditionally-independent of the high-level latent behavior $z_\omega$ given its low-level latent $z_\alpha^i$ (see Appendix A.1.1 for discussion). Next, given initial state distribution $p(s_0)$ and latent behavior spaces, the probability of a trajectory $\tau$ under joint policy $\pi(\cdot)$ is as follows:

$$p^\pi(\tau) = p(s_0) \prod_{t=0}^{T-1} p(s_{t+1}|s_t, a_t) \pi(a_t|s_t) \tag{3}$$

$$= \int_{z_\alpha, z_\omega} p(s_0) \prod_{t=0}^{T-1} p(s_{t+1}|s_t, a_t) \prod_{i=1}^N \pi^i(a_t^i|s_t, z_\alpha^i) p(z_\alpha^i|z_\omega) p(z_\omega) dz_\alpha dz_\omega . \tag{4}$$

We seek to learn the distributions over variables $z_\alpha$ and $z_\omega$, alongside the latent-conditioned policies $\pi^i(a^i|s_t, z_\alpha^i)$, which maximize trajectory probabilities (4). In Appendix A.1, we derive the following variational lower bound, enabling approximation of these components using parametric models:

$$J_{lb} = \mathbb{E}_{\tau \sim \mathcal{D}, z_\alpha \sim q_\phi(z_\alpha|\tau)} \left[ \sum_{t,i} \log \pi_\theta^i(a_t^i|s_t, z_\alpha^i) \right] \tag{5}$$

$$- \beta \left[ \mathbb{E}_{\tau \sim \mathcal{D}, z_\omega \sim q_\phi(z_\omega|\tau)} \left[ \sum_i D_{\mathrm{KL}}(q_\phi(z_\alpha^i|\tau) || p_\theta(z_\alpha^i|z_\omega)) \right] \right. \tag{6}$$

$$\left. + \mathbb{E}_{\tau \sim \mathcal{D}} \left[ D_{\mathrm{KL}}(q_\phi(z_\omega|\tau) || p_\theta(z_\omega)) \right] \right], \tag{7}$$

where $q_\phi(z_\alpha^i|\tau)$ and $p_\theta(z_\alpha^i|z_\omega)$ are, respectively, learned encoder (posterior) and prior distributions over the local behavior latents $z_\alpha^i$; likewise, $q_\phi(z_\omega|\tau)$ and $p_\theta(z_\omega)$ are, respectively, learned encoder and prior distributions over the joint behavior latent $z_\omega$; $\beta$ is a KL-weighting term as in $\beta$-VAEs [72].

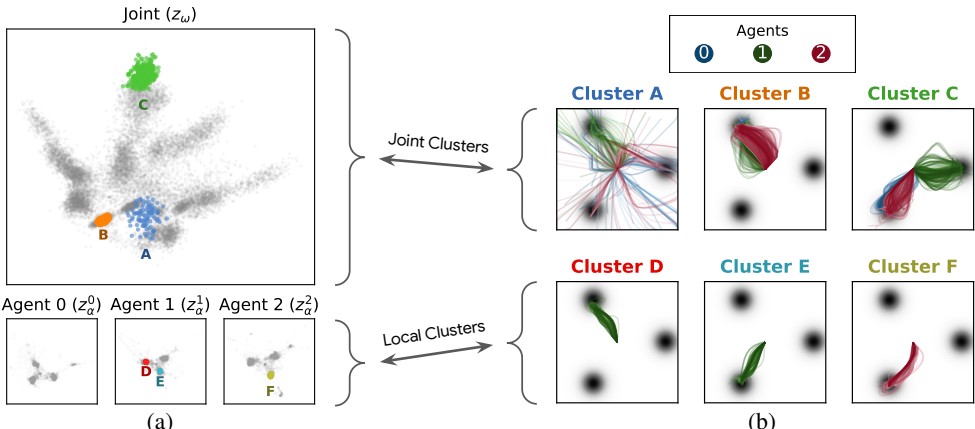

Figure 3: Results for 3-agent hill climbing domain (see interactive version here). (a) Example behavioral clusters discovered by MOHBA. (b) Trajectories corresponding to each cluster, with reward-hills shown in grey. Clusters A to C show joint behaviors, whereas D to F separately show local agent behaviors (with other agents faded in local trajectory plots for readability).

Figure 2b illustrates MOHBA's model architecture, which is informed by the three bound components (5) to (7). During training, each trajectory $\tau \sim \mathcal{D}$ is simultaneously passed through the joint and local encoders, which respectively produce parameters for distributions over $z_\omega$ and $z_\alpha$ (e.g., parameters of Gaussian distributions). Samples of low-level latent vectors $z_\alpha$ are passed to the reconstructed agent policies $\pi_\theta^i(a_t^i | s_t, z_\alpha^i)$, which are trained via the reconstruction component (5). The local KL-divergence component (6) induces the local encoder distribution (which is conditioned directly on $\tau$) to be similar to the local prior distribution (which is conditioned only on samples $z_\omega$), thus enabling meaningful correlations between the encoded local and joint latent space, as later shown. Finally, the joint KL-divergence component (7) is akin to that in a standard variational autoencoder [73, 74]. Overall, MOHBA enables learning of a hierarchical behavioral space (at the joint and local agent levels, $z_\omega$ and $z_\alpha$, respectively) that exposes interesting behavioral clusters.

## 4 Experiments

We showcase various use-cases for MOHBA in a range of domains including continuous coordination games, multiagent MuJoCo [14], and OpenAI hide-and-seek [10]. Appendix A.2 provides data generation, networks, computation, and hyperparameter details. Appendix A.7 provides pseudocode.

**Data generation.** Multiagent trajectory data is generated for each domain via the Acme RL library [75], using the TD3 algorithm [76] in a decentralized MARL fashion, with datasets managed using RLDS [77]. Trajectories are collected at constant intervals throughout training, which also enables analysis of behavioral emergence. We conduct a wide sweep over random seeds for data generation, yielding a diverse trajectory dataset (see Appendix A.5 for dataset details and statistics).

**MOHBA setup.** To analyze the above data using MOHBA, we use a GMM (Gaussian Mixture Model) for the joint prior, a bidirectional LSTM (long short-term memory network) with GMM head for the joint encoder, an MLP (multi-layer perceptron) with Gaussian head for the local priors, a bidirectional LSTM with Gaussian head for the local encoder, and an MLP for reconstructed policies. GMMs are used for the joint prior and encoder as they produce discernible joint behavior clusters [78], whereas the conditioning of the local prior on $z_\omega$ yields such clusters at the local level with a standard Gaussian head. We use parameter-sharing across local priors, local encoders, and reconstructed policies, as common in multiagent setups [79], with a unique one-hot vector identifier appended to agent-specific network inputs to enable heterogeneity in model outputs.

**Independent analysis of joint ($z_\omega$) and local ($z_\alpha$) behaviors.** We here analyze the hierarchical latent structure learned by MOHBA. Specifically, we highlight differences in agent behaviors as identified at the joint and local levels, respectively, by $z_\omega$ and $z_\alpha$. We first conduct a simple sanity check in a 3-agent hill-climbing domain (our earlier example in Fig. 1b). States and actions correspond, respectively, to 2D positions $(x, y)$ and forces $(\Delta x, \Delta y)$ imparted by each agent for

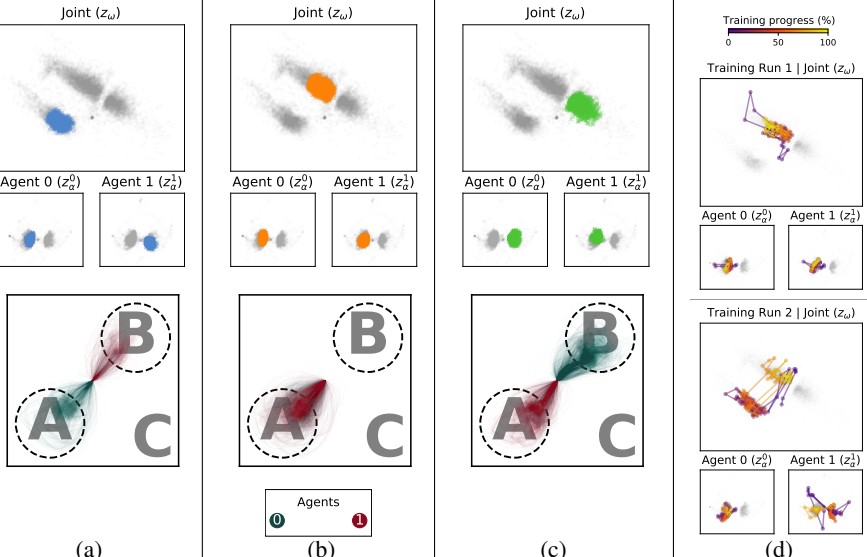

|     | Joint ($z_\omega$) | Joint ($z_\omega$) | Joint ($z_\omega$) | Training progress (%) |
| (a) | (b) | (c) | (d) |

Figure 4: Results for two-agent coordination game (see interactive version here). (a) to (c) show key joint clusters, the coupled local clusters for each agent, and associated trajectories in the domain (over all 50 MARL training runs in the dataset). (d) visualizes the progression of agent behaviors throughout the original MARL training phase, for two example training runs (top and bottom panels).

movement. Agents spawn at the origin and are each rewarded for climbing any of three hills (shown in grey in the domain figures) within the domain in episodes of $50$ timesteps. We generate a dataset using $50$ independent MARL training trials, each conducted for $1e5$ environment steps; trajectories are saved every $200$ steps, yielding $1.25e6$ frames of data ($25000$ trajectories).

Figure 3a highlights several example clusters of joint and local behaviors identified by MOHBA, using samples of $z_\omega$ and $z_\alpha$ from the joint and local encoders. We use Euclidean distances in the original latent spaces to identify nearby vectors, then visualize their 2D projection using principal component analysis. For each cluster, we visualize all associated trajectories in Fig. 3b, thus enabling analysis of behaviors captured in latent space. Clusters A, B, and C correspond to examples of joint behaviors ($z_\omega$). Cluster A contains trajectories early in training, where agents have not learned to converge to a particular hill in the domain. In Cluster B, all agents have learned to navigate towards the top-left hill. In Cluster C, agents 0 and 2 prefer the bottom-left hill, whereas agent 1 prefers the right hill. Overall, clusters at the joint level capture meaningful collective behaviors of the agents.

Next, we analyze individual agent clusters. Note that for all agents, three prominent clusters are apparent in their respective $z_\alpha$ spaces, as each training trial in the data generation process can lead to various agent-wise hill preferences. We compare two such clusters, D and E, for agent 1, observing in the trajectory plots that these corresponds to this agent preferring the top-left and bottom-left hills, respectively. Similarly, Cluster F corresponds to trajectories where agent 2 prefers the bottom-left hill. These results illustrate that the local latents reasonably disentangle each agent's observed behaviors.

**Coupled analysis of joint ($z_\omega$) and local ($z_\alpha$) behaviors.** The above experiments independently analyzed the joint and local latent spaces. We can also concurrently analyze them to better understand local agent contributions to joint behaviors. Consider a two-agent domain with close inter-agent coordination (visualized in the top of Fig. 4a), with state and actions-spaces similar to the hill domain and episodes consisting of

Table 1: Coordination rewards.

|       |   | Agent 1 | | |
|-------|---|---------|--------|--------|
|       |   | $A$ | $B$ | $C$ |
| Agent 0 | $A$ | $(1,1)$ | $(1,1)$ | $(0,0)$ |
|       | $B$ | $(1,1)$ | $(0,0)$ | $(0,0)$ |
|       | $C$ | $(0,0)$ | $(0,0)$ | $(0,0)$ |

$50$ timesteps. Three regions are defined in this domain: $A$ and $B$ (circular regions), and $C$ (region exterior to the circles). When an agent enters a given region, it 'activates' the corresponding strategy in Table 1, with agents receiving rewards at each timestep according to the joint strategy they have activated. For example, if both agents enter region $A$, they each receive a reward of $1$, whereas if one agent enters $A$ while the other is in exterior region $C$, neither receives a reward. This domain involves a significant degree of coordination as agents must discover the rewarding regions, while receiving a sparse reward signal until a valid combination of strategies is discovered. There is also

potential for miscoordination: navigating to region $B$ is rewarding assuming the other agent navigates to $A$, but yields 0 reward if the other agent instead navigates to $B$ (potentially destabilizing training). We run 50 independent MARL sweeps in this domain, yielding $1e6$ data frames (20000 trajectories). As shown in Fig. 4, MOHBA discovers three dominant joint behavior clusters $z_\omega$ here. In each of Figs. 4a to 4c, we highlight one of these clusters and its corresponding local behavior latents $z_\alpha$ for each agent. MOHBA reveals that across the dataset, the agents have learned to cover all 3 optimal joint behaviors in Table 1: $(A, A)$, $(A, B)$, and $(B, A)$. Moreover, despite each agent discovering two local behaviors ($A$ and $B$), $z_\omega$ highlights only the 3 observed joint behaviors (i.e., does not simply highlight 4 clusters consisting of the Cartesian product of individual agents' behavior spaces).

**Behavior emergence throughout MARL training.** We next use MOHBA's latent space to inspect behavior emergence *during* MARL training. Figure 4d visualizes the training progression of two MARL runs used for data generation, tracking behavior changes throughout. In Training Run 1 (top panel of Fig. 4d), the agents converge to and maintain a fixed joint behavior throughout training. By contrast, Training Run 2 (bottom panel) has numerous behavior changepoints, where agents flip back and forth between preferring one of two clusters in $z_\omega$. Concurrently inspecting the $z_\alpha$ space for Training Run 2, we observe that agent 0 converges to and maintains a consistent behavior, whereas agent 1 changes its preference sporadically, also explaining the detected changes in joint behaviors.

**Baseline comparisons.** We next compare against baselines previously used for multiagent behavioral analysis in the literature [9, 11]. An LSTM baseline conducts next-action prediction at each step in a trajectory $\tau$, using an action-prediction loss (APL), defined as the $\mathcal{L}_2$ loss over predicted vs. ground truth actions. This baseline targets using the LSTM hidden states, rather than a learned distribution over latent variables, for understanding and clustering agent behaviors (akin to the analysis in [9, 11]). A flat-VAE baseline provides a non-hierarchical ablation of MOHBA that simply feeds the joint latent $z_\omega$ (rather than $z_\alpha^i$) to reconstructed agent policies (i.e., $\pi_\theta^i(a^i|s, z_\omega)$); we use our usual loss (with the local KL term (6) removed) to train this VAE. We conduct a hyperparameter sweep for the baselines (see Appendix A.2), reporting the best results averaged over 3 random seeds.

Comparisons are conducted at the joint behavior level: for the LSTM, we use the final hidden state as an encoding of each trajectory; for our method and the VAE, we use $z_\omega$ directly. Table 2 compares the methods in the hill-

Table 2: Baseline comparisons. Action-prediction loss (APL) and intra-cluster trajectory distance (ICTD); lower is better for both.

| | Hill-climbing domain | | Coordination game | |
|---|---|---|---|---|
| | APL | ICTD | APL | ICTD |
| LSTM | $4.8 \pm 0.1$ | $0.47 \pm 0.19$ | $2.9 \pm 0.1$ | $0.34 \pm 0.15$ |
| VAE | $6.9 \pm 0.4$ | $\mathbf{0.18 \pm 0.13}$ | $5.0 \pm 0.3$ | $\mathbf{0.16 \pm 0.10}$ |
| MOHBA | $\mathbf{1.3 \pm 0.1}$ | $\mathbf{0.17 \pm 0.13}$ | $\mathbf{2.4 \pm 0.3}$ | $\mathbf{0.16 \pm 0.10}$ |

climbing and coordination game domains. Here we note the APL, which measures how well each method reconstructs ground truth policies. Additionally, we use K-means (sweeping over the # of clusters) to identify behavior clusters for each method, then report the intra-cluster trajectory distance (ICTD), defined as the average distance of all trajectories in a cluster from the mean trajectory in said cluster (akin to intra-cluster point scatter, a common cluster analysis statistic [80]) . Combined, these measures provide a proxy for evaluating the latent representations in terms of enabling accurate reconstructions (APL) while clustering similar trajectories together (ICTD). MOHBA significantly outperforms the LSTM and VAE baselines in terms of APL, while also clustering similar trajectories at the $z_\omega$ level in terms of ICTD. We provide additional results including APL throughout training, full ICTD sweeps, and visualizations of the latent spaces for the baselines in Appendix A.3.2.

**Behavior concept discovery.** Next, we test the representation power of the latent spaces learned by MOHBA, and illustrate a means of discovering 'behavior concepts' in the latent space. We adopt the completeness-aware concept explanations framework of Yeh et al. [13], with slight changes to make it amenable to this setting (see Appendix A.2.5). At a high-level, given a set of inputs, a set of concept vectors $C$ in the same space as inputs, and prediction targets (e.g., domain characteristics of interest), Yeh et al. [13] define a framework to compute class-conditioned Shapley values, called ConceptSHAP. To do so, every input is projected onto each of the concept vectors, yielding a vector of 'concept scores', which is then passed to a simple prediction head $g$ (e.g., small MLP or linear model) to predict the targets and compute the ConceptSHAP. ConceptSHAP provides numerical scores interpreted as the importance of each concept in $C$ for predicting a given target class, which is useful for identifying key concepts (and nearby inputs) associated with certain domain characteristics.

Table 3: Discovered concepts using $z_\omega$ in hill-climbing domain. For each characteristic (agent dispersion and return), 5 classes are constructed using the ground truth information in trajectories $\tau$. The concept explanation framework of Yeh et al. [13] is then used to first predict the correct classes (using only $z_\omega$, rather than the trajectory $\tau$, as input), then identify core concepts related to each class.

| Concept | Top discovered concept using $z_\omega$ (per class, with average concept measure beneath) | | | | |
|---|---|---|---|---|---|
| | Class 0 | Class 1 | Class 2 | Class 3 | Class 4 |
| Agent Dispersion (Classification accuracy: 54.27%) | $0.05 \pm 0.13$ | $0.15 \pm 0.06$ | $1.79 \pm 0.10$ | $2.18 \pm 0.19$ | $2.36 \pm 0.10$ |
| Total Return (Classification accuracy: 60.98%) | $361.0 \pm 256.0$ | $671.4 \pm 115.1$ | $611.1 \pm 244.2$ | $764.9 \pm 72.4$ | $820.4 \pm 155.0$ |

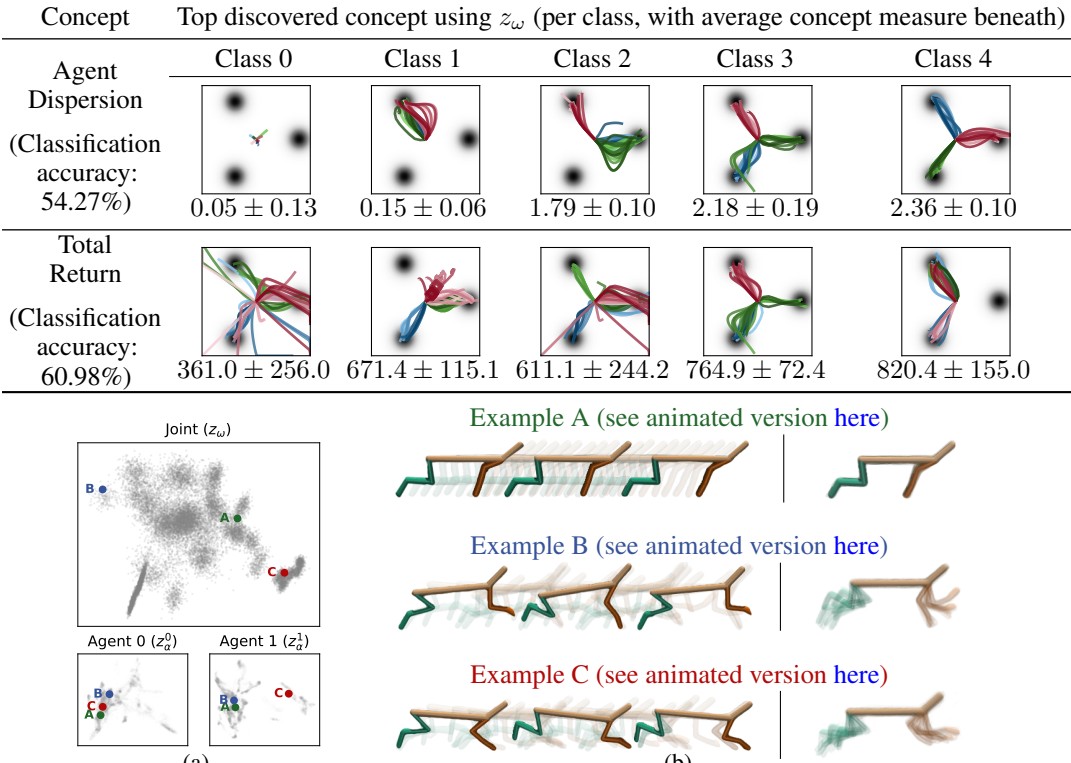

Figure 5: Multiagent MuJoCo HalfCheetah behavioral space (see interactive version here). Agent 0 and 1, respectively, control the back and front limbs, coordinating to move the cheetah. (a) Detected behavioral space, with three example trajectories indicated. (b) HalfCheetah behavior corresponding to the same three examples (left panels showing trajectory frames, and right panels showing the same frames with the cheetah torso aligned to disambiguate the back and front leg agent behaviors).

In Table 3, we use this technique to identify concepts in $z_\omega$ space associated with varying characteristics in the hill-climbing domain. The concept set $C$ considered is generated using K-means in the $z_\omega$ space (see Appendix A.2.5 for details). The first row of Table 3 shows classes corresponding to increasing levels of agent dispersion in the domain (with dispersion defined as the sum of $\mathcal{L}_2$-distances of agents from their centroid at the final timestep). For each trajectory $\tau$, we compute the associated dispersion, creating 5 classes of equal-sized bins (labels 0 through 4 mapping to bins of increasing dispersion). We create an 80-20 train-validation split, then train a 2-layer (8 hidden units each) MLP $g$ via a softmax-cross entropy loss to predict the classes using only $z_\omega$ as input (rather than the actual trajectory $\tau$). We attain a validation accuracy of 54.27%, signifying the predictive capabilities of $z_\omega$. Next, we compute the class-conditioned ConceptSHAP, thus identifying the top-scoring concept vector for each class. In Table 3, we visualize the 20 trajectories with the closest $z_\omega$ to the top-scoring concept for each class (with the mean agent dispersion across these trajectories listed under each image). Intuitively, the identified concepts involve agents becoming increasingly dispersed, with agents first nearly stationary at the center (class 0), then all converging to the same hill (class 1), spreading across two hills (class 2), and finally covering all hills (classes 3 and 4).

The second row of Table 3 repeats this experiment, now using classes associated with agents' sum-of-returns. We attain a validation set accuracy of 60.98%, and observe behaviors associated with generally increasing reward by identifying top clusters in each class (with some overlaps, e.g., classes 1 and 2 with high standard deviation in returns). Overall, these experiments help quantify the representational capacity of $z_\omega$ and identify clusters associated with distinct behavioral characteristics.

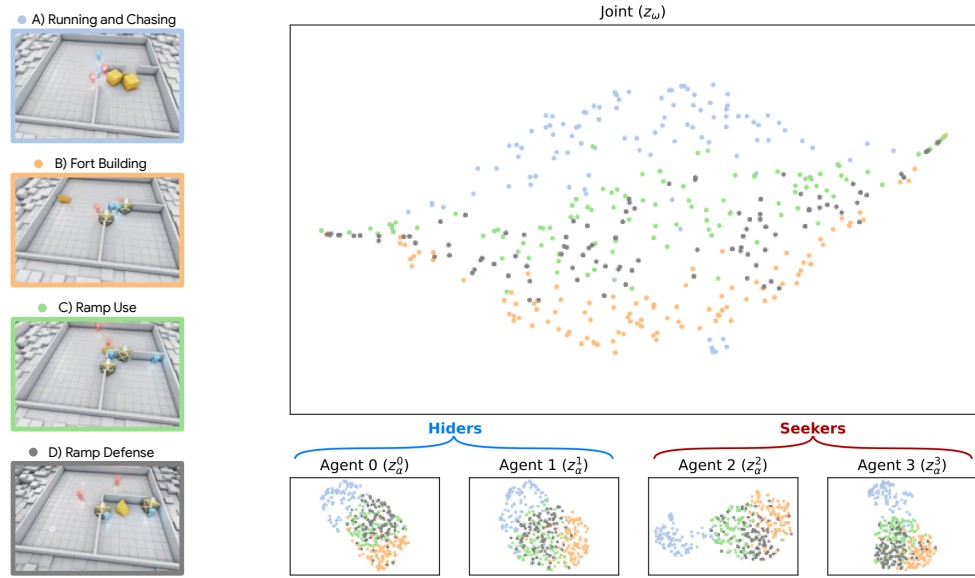

Figure 6: Results for the OpenAI hide-and-seek environment [10], involving two teams of agents (2 hiders and 2 seekers). We mix the trajectories collected from 4 OpenAI-annotated behavioral checkpoints (indicated on the left panel) together, then train MOHBA on this shuffled dataset. Using MOHBA, we observe the presence of behavior clusters that correspond well to human-annotated policy labels, both at the joint and local agent levels, despite our algorithm not having access to policy labels during training. Domain screenshots reproduced with permission from Baker et al. [10].

**Scalability to high-dimensional domains.**    We next test MOHBA's scalability to high-dimensional domains using the 2-Agent HalfCheetah multiagent MuJoCo domain [14], where agents 0 and 1, respectively, control the back and front limbs of a cheetah to coordinate movement. Each agent's state consists of a 6D vector summarizing velocity and position information for the 3 joints it controls, with its 3D action space corresponding to motor torques applied to these joints. The agents coordinate to maximize the forward-speed of the cheetah over episodes of 200 timesteps each. We generate data using 30 independent trials of MARL training, collecting $1e5$ total trajectories to train MOHBA.

Figure 5a visualizes the behavior space learned by MOHBA, wherein we observe several clusters. We highlight three example trajectories stemming from distinct clusters in the joint behavior space $z_\omega$, showing the corresponding HalfCheetah behaviors in Fig. 5b. The left panel of each example provides a view of the cheetah's overall movement, where we observe key behavior differences: in Example A, the cheetah runs forward using subtle vibration of its limbs; in Example B, it bounces forward with its torso arched up due to its front limb (agent 1) being more extended than its back limb (agent 0); in Example C, it moves closer to the ground with its torso arched down. In the latent space (Fig. 5a), we observe that for Examples B and C, the $z_\alpha^0$ (back limb) latents are close to one another, while the $z_\alpha^1$ (front limb) latents are far apart. To investigate these local agent behavioral differences, the right panel of Fig. 5b provides an overlaid view of the same frames, with the torso now aligned (making it easier to discern the behaviors of agent 0 vs. agent 1). Here we observe that the back limb (agent 0) behaves similarly across both examples, while the front limb (agent 1) stays much closer to the head for Example C, in contrast to Example B, which coincides with the findings in the $z_\alpha$ space. Appendix A.3.1 provides additional results for a 4-agent MuJoCo AntWalker environment.

**Application to externally-trained policies: hide-and-seek game.**    We next apply MOHBA to policies trained by external teams. We consider the OpenAI hide-and-seek environment [10], where 2 hiders and 2 seekers compete in a rich environment with various interactive objects (boxes and ramps). OpenAI has open-sourced policies [81] annotated by humans as exhibiting distinctive behaviors at key stages of training. We consider four policy checkpoints that correspond to the following human annotations: A) 'running and chasing', B) 'fort building', C) 'ramp use', and D) 'ramp defense'. The state-space used for each of the agents is 100-dimensional, consisting of the agent's own state (position, rotation, and velocity), states of the other 3 agents, and the states of 3 boxes and 1 ramp in the environment (position, velocity, and box size); each agent's action space consists of a

3-dimensional force vector, and a 'glue' and 'lock' action for interacting with objects. We collect 100 trajectories per policy checkpoint, each being 200 timesteps long. These trajectories exhibit a wide distribution of behaviors, as agent and object initializations are random in each episode. We then mix the trajectories collected from all policy checkpoints, then train MOHBA on this shuffled dataset.

Figure 6 visualizes the behavior spaces discovered by MOHBA in the hide-and-seek domain. Agents 0 and 1 in this figure correspond to 'hiders', whereas agents 2 and 3 are 'seekers'. We label each of the trajectories in this figure with the human-expert annotations provided by OpenAI. In Fig. 6, we observe the presence of behavior clusters that correspond well to the human-expert labels, both at the joint and local agent levels. Interestingly, for the seekers (agents 2 and 3), policy A ('running and chasing') is highly distinctive and well-separated from the other behaviors. Moreover, despite the order of emergent behaviors in the original hide-and-seek MARL training being A → B → C → D, policies B ('fort building') and D ('ramp defense') appear to be behaviorally slightly closer to one another than C ('ramp use') and D, both in the joint space and also the local spaces of the seekers (agent 2 and 3). This could perhaps be due to both the 'fort building' and 'ramp defense' policies being associated with situations where the seekers cannot easily find the hiders, due to the hiders using obstacles to block entrances (B) and moving ramps to prevent their effective use (D). Overall, these experiments help to validate MOHBA's learned latent spaces using policy labels manually annotated by human experts, and highlight its applicability to behaviorally-rich domains.

## 5 Discussion

Our proposed method, MOHBA, leverages trajectory data to better understand multiagent behaviors during and after training. MOHBA assumes no knowledge of agents' underlying training algorithms, does not require access to their hidden states or internal models, and applies even to reward-free settings. Our experiments showcased a variety of applications of MOHBA, including the analysis of joint and local agent behaviors, monitoring of behavior emergence throughout training, discovery of behavioral concepts associated with certain domain criteria, and disentanglement of third-party-labeled behaviors from open-source policies such as those for OpenAI hide-and-seek [10].

While we believe our approach is an important step in terms of increasing the understanding of multiagent systems, there are several limitations and potential societal impacts of note. One limitation is related to the collection of a dataset of agent behaviors. Understanding emergent behaviors throughout MARL training using MOHBA requires the storage of large-scale trajectory datasets, which could potentially take a lot of storage and compute power to generate. It would be interesting to consider follow-ups to our model that learn behavioral clusters in a streaming fashion, thus building knowledge of behaviors over time and permitting older data to be discarded throughout agent training. Moreover, in our datasets, we collected trajectories at uniform intervals throughout original MARL training. However, it might be interesting to consider a non-uniform collection scheme, e.g., collecting trajectories only when they are detected to diverge from behavioral clusters of a pre-trained MOHBA model. Finally, it might be interesting to enable the agents to first learn joint behaviors / 'skills' using the proposed method, and leverage them to transfer to related downstream tasks (potentially more sample-efficiently than training from scratch).

The study of behavioral interactions in multiagent systems can potentially be used for both positive and negative societal applications. For example, such an approach could be used to prevent certain undesirable behaviors by agents that interact with humans (e.g, self-driving cars), but potentially also used by adversaries to predict and exploit other behaviors (e.g., exploiting certain human preferences for harm), or even inadvertently cause harm due to misinterpretation of certain behavioral modes. As such, further research and evaluation will be required prior to deployment of this and related behavioral analysis approaches to human-facing domains. Nonetheless, as agent capabilities continue to grow, our view is that behavioral analysis of multiagent systems will become increasingly important and should complement traditional reward-based performance monitoring.

## Acknowledgments and Disclosure of Funding

We thank Asma Ghandeharioun, Meredith Morris, and Kathy Meier-Hellstern for their helpful feedback and support during the paper writing phase.

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
