# A  Appendix for 'Beyond Rewards: a Hierarchical Perspective on Offline Multiagent Behavioral Analysis'

## A.1  Derivations

### A.1.1  On the Conditional Independence of $\pi^i$ from $z_\omega$ given $z_\alpha^i$

In (1), we assumed that each agent's latent-conditioned policy is conditionally-independent of the high-level latent behavior $z_\omega$ given its low-level latent $z_\alpha^i$. This section provides justification of this assumption.

One of the prototypical paradigms in multiagent (MARL) training is that of 'centralized training, decentralized execution'. In this regime, each agent's decision-making policy is conditioned only on its available local information / local observations during execution; thus, the behaviors exhibited by each agent are ultimately informed by their local information, rather than global (joint) information.

In our setting, $z_\alpha^i$ serves to provide this local behavior context for each agent. Thus, the assumption of the policy being conditionally-independent of $z_\omega$ given $z_\alpha^i$ corresponds well to the assumption of agents only using local information (rather than joint information) in MARL to inform their policy/decision-making.

Having said this, we also note that there is a strong relationship between local and joint behavioral distributions in our setting, which helps provide a coordination signal to the policy implicitly through $z_\alpha^i$. Specifically, our derivation results in a local behavior prior term, $p(z_\alpha^i|z_\omega)$; using this term, the joint behavior latent $z_\omega$ is able to influence the learned space over local behavior latents $z_\alpha^i$. Thus, the information-flow in our framework can be more intuitively described as follows:

1. The joint behavior latent observes a trajectory and summarizes the joint behavior exhibited in it (e.g., team-wide cooperation, competition, etc.) in $z_\omega$.
2. Subsequently, this joint latent $z_\omega$ affects the local behavior spaces $z_\alpha^i$, which are then sampled and used to inform each agent how to behave locally in order to achieve the joint behavior observed.

Overall, the above relationship between the two latent spaces implies makes the conditional-independence of the policy from $z_\omega$ a reasonable simplifying assumption.

### A.1.2  Derivation of the Variational Lower Bound

This section details the derivation of the variational lower bound described in Section 3 of the main text, which is used for training MOHBA.

Similar to traditional variational autoencoder approaches [73, 74], we approximate the maximization of the latent-conditioned trajectory probability (4) in the main text using the evidence lower bound

$$J_{lb} = \mathbb{E}_{\tau \sim \mathcal{D}, q_\phi(z_\alpha, z_\omega|\tau)} \left[\log p^\pi(\tau|z_\alpha, z_\omega)\right] - \mathbb{E}_{\tau \sim \mathcal{D}}[D_{\mathrm{KL}}(q_\phi(z_\alpha, z_\omega|\tau)||p_\theta(z_\alpha, z_\omega))] , \quad (8)$$

where $q_\phi(\cdot|\tau)$ and $p_\theta(\cdot)$ are, respectively, learned posterior and prior distributions.

Using (4) from the main text to expand this expression yields

$$J_{lb} = \mathbb{E}_{\tau \sim \mathcal{D}, q_\phi(z_\alpha, z_\omega|\tau)} \left[\log p(s_0) + \sum_t \log p(s_{t+1}|s_t, a_t) + \sum_{t,i} \log \pi_\theta^i(a_t^i|s_t, z_\alpha^i)\right]$$
$$- \mathbb{E}_{\tau \sim \mathcal{D}} \left[D_{\mathrm{KL}}(q_\phi(z_\alpha, z_\omega|\tau)||p_\theta(z_\alpha, z_\omega))\right] \quad (9)$$

Note that the terms $p(s_0)$ and $p(s_{t+1}|s_t, a_t)$ stem from the underlying MA-MDP environment, and thus cannot be optimized via parameters $\phi$ and $\theta$. Dropping these extraneous terms yields,

$$J_{lb} = \mathbb{E}_{\tau \sim \mathcal{D}, q_\phi(z_\alpha|\tau)} \left[\sum_{t,i} \log \pi_\theta^i(a_t^i|s_t, z_\alpha^i)\right] - \mathbb{E}_{\tau \sim \mathcal{D}} \left[D_{\mathrm{KL}}(q_\phi(z_\alpha, z_\omega|\tau)||p_\theta(z_\alpha, z_\omega))\right] . \quad (10)$$

The first term simply relates to trajectory reconstruction, inducing our agent-wise policies $\pi_\theta^i$ to behave similarly to the observed trajectories in the dataset. The second term is a regularization term

involving our two latent parameters, which we further simplify using assumptions used in previous multi-level VAE models [67]. First, we assume that the posterior distribution is factorizable when conditioned on a particular trajectory, as follows,

$$q_\phi(z_\alpha, z_\omega | \tau) = q_\phi(z_\alpha | \tau) q_\phi(z_\omega | \tau) \tag{11}$$

$$= \left[ \prod_i q_\phi(z_\alpha^i | \tau) \right] q_\phi(z_\omega | \tau), \tag{12}$$

and similarly for the prior,

$$p_\theta(z_\alpha, z_\omega) = p_\theta(z_\alpha | z_\omega) p_\theta(z_\omega) \tag{13}$$

$$= \left[ \prod_i p_\theta(z_\alpha^i | z_\omega) \right] p_\theta(z_\omega). \tag{14}$$

We next simplify the KL-divergence component of (10) as follows by combining it with (12) and (14):

$$D_{\mathrm{KL}}(q_\phi(z_\alpha, z_\omega | \tau) || p_\theta(z_\alpha, z_\omega)) \tag{15}$$

$$= \int_{z_\alpha, z_\omega} q_\phi(z_\omega, z_\alpha | \tau) \log \frac{q_\phi(z_\alpha, z_\omega | \tau)}{p_\theta(z_\alpha, z_\omega)} d_{z_\alpha} d_{z_\omega} \tag{16}$$

$$= \int_{z_\alpha, z_\omega} q_\phi(z_\alpha | \tau) q_\phi(z_\omega | \tau) \log \frac{q_\phi(z_\alpha | \tau) q_\phi(z_\omega | \tau)}{p_\theta(z_\alpha | z_\omega) p_\theta(z_\omega)} d_{z_\alpha} d_{z_\omega} \tag{17}$$

$$= \int_{z_\alpha, z_\omega} \left[ q_\phi(z_\alpha | \tau) q_\phi(z_\omega | \tau) \log \frac{q_\phi(z_\alpha | \tau)}{p_\theta(z_\alpha | z_\omega)} + q_\phi(z_\alpha | \tau) q_\phi(z_\omega | \tau) \log \frac{q_\phi(z_\omega | \tau)}{p_\theta(z_\omega)} \right] d_{z_\alpha} d_{z_\omega} \tag{18}$$

$$= \int_{z_\alpha, z_\omega} q_\phi(z_\omega | \tau) \prod_j q_\phi(z_\alpha^j | \tau) \sum_i \log \frac{q_\phi(z_\alpha^i | \tau)}{p_\theta(z_\alpha^i | z_\omega)} d_{z_\alpha} d_{z_\omega}$$
$$+ \int_{z_\omega} q_\phi(z_\omega | \tau) \log \frac{q_\phi(z_\omega | \tau)}{p_\theta(z_\omega)} d_{z_\omega} \underbrace{\int_{z_\alpha} q_\phi(z_\alpha | \tau) d_{z_\alpha}}_{=1} \tag{19}$$

$$= \sum_i \int_{z_\alpha, z_\omega} q_\phi(z_\omega | \tau) \left( \prod_j q_\phi(z_\alpha^j | \tau) \right) \log \frac{q_\phi(z_\alpha^i | \tau)}{p_\theta(z_\alpha^i | z_\omega)} d_{z_\alpha} d_{z_\omega} + D_{\mathrm{KL}}(q_\phi(z_\omega | \tau) || p_\theta(z_\omega)) \tag{20}$$

$$= \sum_i \int_{z_\alpha^i, z_\omega} q_\phi(z_\omega | \tau) q_\phi(z_\alpha^i | \tau) \log \frac{q_\phi(z_\alpha^i | \tau)}{p_\theta(z_\alpha^i | z_\omega)} d_{z_\alpha^i} d_{z_\omega} \left[ \prod_{j \neq i} \underbrace{\int_{z_\alpha^j} q_\phi(z_\alpha^j | \tau) d_{z_\alpha^j}}_{=1} \right]$$
$$+ D_{\mathrm{KL}}(q_\phi(z_\omega | \tau) || p_\theta(z_\omega)) \tag{21}$$

$$= \mathbb{E}_{q_\phi(z_\omega | \tau)} \left[ \sum_i D_{\mathrm{KL}}(q_\phi(z_\alpha^i | \tau) || p_\theta(z_\alpha^i | z_\omega)) \right] + D_{\mathrm{KL}}(q_\phi(z_\omega | \tau) || p_\theta(z_\omega)) \tag{22}$$

Combining this result with (10) and additionally modulating the KL-terms in the lower bound as in $\beta$-VAEs [72], our overall objective simplifies to

$$J_{lb} = \mathbb{E}_{\tau \sim \mathcal{D}, z_\alpha \sim q_\phi(z_\alpha | \tau)} \left[ \sum_{t,i} \log \pi_\theta^i(a_t^i | s_t, z_\alpha^i) \right] \tag{23}$$

$$- \beta \left[ \mathbb{E}_{\tau \sim \mathcal{D}, z_\omega \sim q_\phi(z_\omega | \tau)} \left[ \sum_i D_{\mathrm{KL}}(q_\phi(z_\alpha^i | \tau) || p_\theta(z_\alpha^i | z_\omega)) \right] \right. \tag{24}$$

$$\left. + \mathbb{E}_{\tau \sim \mathcal{D}} [D_{\mathrm{KL}}(q_\phi(z_\omega | \tau) || p_\theta(z_\omega))] \right]. \tag{25}$$

## A.2 Experiment Details and Hyperparameters

This section provides an overview of the details for our experiments.

### A.2.1 Data Generation Details

The multiagent trajectory data analyzed in this paper is generated using the Acme RL library [75], using the TD3 algorithm [76] in a decentralized fashion, with dataset management handled using RLDS [77]. Each agent uses a 2-layer MLP (256 hidden units for each layer) for its TD3 network (using the vanilla TD3 network specifications in Acme), with the hyperparameters used for data generation summarized for each domain in Table 4. Note that the large number of training trials/seeds used in this data generation pipeline serve to produce a wide variety of agent behaviors in the dataset subsequently analyzed by MOHBA.

Table 4: Hyperparameters used for data generation. Values in braces indicate hyperparameter sweeps.

| Parameter | Hill-climbing | Coordination Game | HalfCheetah | AntWalker |
|---|---|---|---|---|
| MARL algorithms used | TD3 | TD3 | TD3 | {TD3, SAC} |
| # agents | 3 | 2 | 2 | 4 |
| Episode steps | 50 | 50 | 200 | 300 |
| # steps per training trial | 1e5 | 2e5 | 2e5 | 2e5 |
| TD3 / SAC batch size | 256 | 256 | 256 | {32, 256} |
| TD3 / SAC learning rates | 5e-5 | 5e-5 | 5e-4 | {1e-3 to 9e-3} |
| TD3 $\sigma$ | 0.1 | 0.1 | 0.1 | 0.1 |
| TD3 target $\sigma$ | 0.1 | 0.1 | 0.1 | 0.1 |
| TD3 $\tau$ | 0.005 | 0.005 | 0.005 | 0.005 |
| TD3 delay | 2 | 2 | 2 | 2 |
| # seeds | 50 | 50 | 30 | 50 |

### A.2.2 MOHBA Hyperparameters

We conduct a wide hyperparameter sweep for training MOHBA itself, summarized in Table 5. Our joint and local encoder models each consist of a bidirectional LSTM, followed by a 2-layer MLP head for mapping to latent distribution parameters; a GMM head is used for the joint encoder, and a Gaussian for the local encoder. Our local prior model consists of a 2-layer MLP with Gaussian head, with our joint prior model simply being the learned parameters of a GMM. All models use ReLU for intermediate layer activations. Note that we found that cyclically-annealing [82] the $\beta$ term in our variational lower bound from 0 to the values specified in Table 5 to help avoid KL-vanishing. Sweeps are conducted over the MOHBA latent space dimensionality (4 and 8-dimensional), MLP hidden sizes (64 and 128 units), LSTM hidden sizes (64 and 128 units), Adam optimizer [83] learning rates (0.001 and 0.0001), with 3 seeds per parameter set (standard deviations are reported over all seeds). All MOHBA and baseline training is conducted using the Adam optimizer [83], with Adam parameters $\beta_1 = 0.9$, $\beta_2 = 0.999$, gradient clipping using a max global norm threshold of 10.0, and learning rates swept over as indicated in the tables.

### A.2.3 Baseline Model Hyperparameters

We similarly conduct a wide sweep over hyperparameters for the considered baselines, summarized in Table 6.

### A.2.4 Computational Details

For MARL trajectory data generation, we used an internal CPU cluster for both the 3-agent hill-climbing and 2-agent coordination domains, using TPUs for only the multiagent MuJoCo data generation. For training MOHBA itself, we used TPUs for the 3-agent hill-climbing and 2-agent coordination domains, and a Tesla V100 GPU cluster for the MuJoCo environment. For training the LSTM and VAE baselines, we used Tesla P100 GPU clusters.

Table 5: Hyperparameter sweeps for training MOHBA. Values swept over are indicated via braces.

| Parameter | Value |
|---|---|
| Training steps | 1e5 |
| Batch size (# trajectories) | 128 |
| Latent $z_\omega$ and $z_\alpha^i$ dimensionality | {4, 8} |
| Hidden units (joint encoder/local prior/local encoder MLPs) | {64, 128} |
| Hidden units (joint encoder /local encoder LSTMs) | {64, 128} |
| Hidden units (reconstructed policy MLP) | 32 |
| GMM mixture size (joint prior/joint encoder) | 8 |
| Adam optimizer learning rates | {1e-3, 1e-4} |
| KL-loss $\beta$ weighing term | {1e-4, 1e-2} |
| KL-loss cyclical annealing period | {5e3, 1e4} |
| # seeds | 3 |

Table 6: Hyperparameter sweeps for baselines. Values swept over are indicated via braces.

| Parameter | LSTM Baseline | VAE Baseline |
|---|---|---|
| Training steps | 1e5 | 1e5 |
| Batch size (# trajectories) | 128 | 128 |
| Latent $z_\omega$ dimensionality | N/A | {4, 8} |
| Hidden units (joint encoder/local prior/local encoder MLPs) | N/A | {64, 128} |
| Hidden units (joint encoder /local encoder LSTMs) | {64, 128} | {64, 128} |
| Hidden units (reconstructed policy MLP) | 32 | 32 |
| GMM mixture size (joint prior/joint encoder) | N/A | 8 |
| Adam optimizer learning rates | {1e-3, 1e-4} | {1e-3, 1e-4} |
| KL-loss $\beta$ | N/A | {1e-4, 1e-2} |
| KL-loss cyclical annealing period | N/A | {5e3, 1e4} |
| # seeds | 3 | 3 |

### A.2.5 Concept Discovery Framework

This section describes details of how we apply the completeness-aware concept-based explanation framework of Yeh et al. [13] to discover interesting concepts in our setting.

Given a characteristic of interest (e.g., the level of dispersion of agents), we define a training set consisting of joint latents $z_\omega$ and class labels $y$ (e.g., classes corresponding to different intervals of team returns). Yeh et al. [13] seek to identify a set of concept vectors that are sufficient for predicting the labels given the inputs. Specifically, they let $C = \{c_j\}_{j=1}^m$ denote the set of concepts, which are unique vectors where $c_j \in \mathbb{R}^{D_\omega} \forall j$. Normalizing $z_\omega$, we can then use the inner product $\langle z_\omega, c_j \rangle$ as a similarity measure between $z_\omega$ and concept $c_j$. The *concept product* is defined $\nu_{\mathbf{c}}(z_\omega) = \mathrm{TH}(\langle z_\omega, c_j \rangle, \kappa) \in \mathbb{R}^m$, where $\mathrm{TH}(\cdot, \kappa)$ clips values less than $\kappa$ to 0. Normalizing the concept product yields the *concept score* $\hat{\nu}_{\mathbf{c}}(z_\omega) = \nu_{\mathbf{c}}(z_\omega)/||\nu_{\mathbf{c}}(z_\omega)||_2 \in \mathbb{R}^m$, where elements provide a measure of similarity of the input $z_\omega$ to each of the $m$ concept vectors.

Using these definitions, we can gauge the representational power of $z_\omega$ by learning a mapping $g : \hat{\nu}_{\mathbf{c}}(z_\omega) \rightarrow y$. In practice, $g$ is a simple model (e.g., shallow network or linear projection) so as to gauge the expressivity of the latent space. Given a mapping $g$, we use the classification accuracy as a 'completeness score' [13] for the set of concepts $C$, defined $\eta = \sup_g P_{z_\omega, y \sim V} \left[ y = \arg\max_{y'} g(\nu_{\mathbf{c}}(z_\omega)) \right]$, where $V$ is a validation set. Importantly, this approach permits us to compute class-conditioned Shapley values, called ConceptSHAP in the framework of [13]. Specifically, for a given class $k$, the class-conditioned ConceptSHAP value for each concept $c_j$ is defined,

$$\lambda_j(\eta_k) = \sum_{S \subseteq C \backslash c_j} = \frac{(m - |S| - 1)!|S|!}{m!} \left[ \eta_k(S \cup \{c_j\}) - \eta_k(S) \right], \quad (26)$$

where $\eta_k$ is the completeness score for class $k$ (computed simply as the classification accuracy of the model for the subset of validation points with ground truth label $k$). ConceptSHAP provides a

measure of importance of each concept $c_j$ for predicting the outcomes associated with a given class, which we can then use to identify the trajectories. The experiments in our main paper use the above setup to both quantitatively evaluate $z_\omega$ in terms of representation power, and also reveal interesting concepts associated with relevant characteristics in the domain.

For each of the prediction problems considered in the main paper (agent dispersion and agent returns), we create an 80-20 training-validation data split (over 5 classes). For concept generation we use K-means with 16 and 24 clusters, a 2-layer MLP (with 8 hidden units per layer) for the prediction head $g$, training with a batch size of $64$, and a concept threshold $\kappa$ of 0.0 and 0.3 for the dispersion and agent return experiments, respectively, and train the prediction head for $1e4$ steps. As in Yeh et al. [13], we use KernelSHAP [84] to approximate ConceptSHAP efficiently.

### A.3 Additional Results

#### A.3.1 Additional Large-scale Experiments

**MultiAgent MuJoCo AntWalker Domain Results.** We also consider the MultiAgent MuJoCo AntWalker domain, wherein 4 agents each control one of 4 ant legs to coordinate movement towards the $+x$ direction (Fig. 7a). To collect data for this domain, we conduct a wider MARL parameter sweep using both the TD3 and SAC algorithms, with varying training batch sizes and learning rates to gather a widely-varying dataset of behaviors. We subsequently train MOHBA on this data, visualizing the learned behavior clusters in Fig. 7b. We observe several behavior clusters of interest; notably, a large joint cluster exists in the top-right region of the joint and local returns, which, upon inspection of underlying trajectory videos, corresponds to cases where agents attain very low return. Similarly, there exists a smaller cluster in the lower-left region of the joint returns that also attains very low performance. The remaining clusters correspond to the agents displaying various ant-poses, and moving only incrementally. One exceptional cluster also exists in the left region of the joint behavior space, which attains medium-level return (points that are primarily red in color). On closer inspection, the AntWalker behavior in this cluster corresponds to one of the agents learning a reasonably good walking gait, while the remaining three agents remain stationary.

#### A.3.2 Additional Baseline Results

Figures 8 and 9 provide additional results for the baseline comparisons conducted in the main paper. Specifically, (a) in each figure provides an expanded comparison of ICTD (with a sweep over the # of clusters in K-means). In (b) of each figure, we visualize the convergence of the APL throughout training for MOHBA and the baselines. Subfigures (c) and (d), respectively, show a PCA and UMAP projection of the latent space discovered by the LSTM baseline (recall from the main paper that 'joint embeddings' used for the LSTM are simply the final hidden state). Likewise, subfigures (e) and (f) show the same projections for the VAE baseline.

As especially evident in the coordination domain (Fig. 9), the joint LSTM baseline embeddings (whether using PCA or UMAP projections, Figs. 9c and 9d, respectively) do not reveal to the fairly interpretable 3 clusters of coordinated behaviors discussed in the main paper. By contrast, the VAE baseline (in both the PCA and UMAP case, Figs. 9e and 9f, respectively) does produce these clusters; despite this, note that the key limitation of this baseline is its non-hierarchical nature, which makes identification of the local-to-joint behavioral correspondences noted in the main paper significantly more difficult.

### A.4 Hyperparameter Ablations and Effects on Latent Spaces

This section conducts an analysis of the effect of key hyperparameters on the latent spaces learned by MOHBA. We focus our analysis on the KL-loss $\beta$ term in the training objective, alongside the latent $z_\omega$ and $z_\alpha$ dimensionality. We both compare the policy reconstruction loss (5) across ablations to attain a quantitative comparison of the quality of the latent encodings, and additionally compare the structure of the learned latent spaces themselves as a function of these parameters for a qualitative understanding.

First, in terms of the raw policy reconstruction loss, we find in Figs. 10a, 11a and 12a that the latent dimensionality has a higher impact on policy reconstruction performance than the weighing term $\beta$; intuitively, increasing latent dimensionality leads to better reconstruction (lower loss) due to the latent

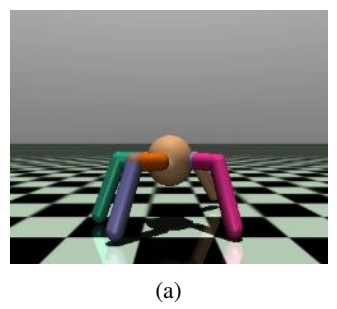

(a)

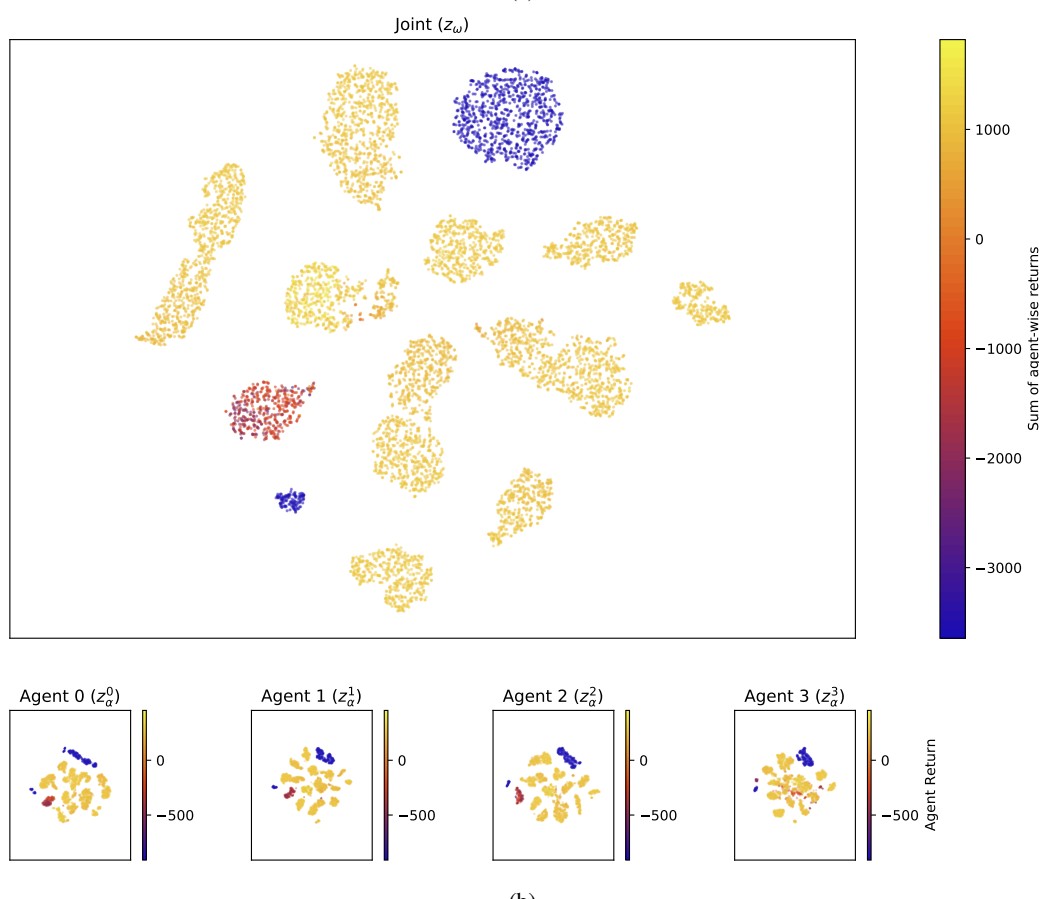

(b)

Figure 7: Results for Multiagent MuJoCo AntWalker domain. (see interactive version here). (a) This domain involves 4 agents, each controlling one of the ant legs to coordinate movement towards the $+x$ direction. (b) Behavior space learned by MOHBA in the AntWalker domain.

space being able to encode more behavioral information about the agent trajectories. By contrast, the effects of the KL weighing term are more negligible in terms of reconstruction loss.

Second, we inspect the effects of these hyperparameter sweeps on the learned latent distributions themselves. Specifically, the respective panels (b)-(e) of each of Fig. 10, Fig. 11, and Fig. 12 visualize the change in latent space structure as a function of the latent dimensionality and $\beta$. At a high-level, prominent behavioral clusters mentioned in the main text are re-discovered throughout these parameter sweeps (e.g., the three local clusters for the hill climbing environment, the joint and local clusters for the coordination environment, and the various walking gait clusters for the HalfCheetah environment). For the more complex latent spaces (e.g., HalfCheetah), increasing the dimensionality of $z$ tends to increase the number of joint clusters identified (e.g., compare Fig. 12c versus Fig. 12e); this is intuitive as a larger latent space results in a richer encoding, capable of distinguishing more nuanced behaviors. Increasing the KL $\beta$ term from $1e-4$ to $1e-2$ tends to

|        | APL | ICTD | | |
|--------|-----|------|------|------|
|        |     | K=4 | K=8 | K=16 |
| LSTM   | $4.8 \pm 0.1$ | $0.60 \pm 0.15$ | $0.51 \pm 0.17$ | $0.47 \pm 0.19$ |
| VAE    | $6.9 \pm 0.4$ | $0.44 \pm 0.11$ | $0.30 \pm 0.14$ | $0.18 \pm 0.13$ |
| MOHBA  | $1.3 \pm 0.1$ | $0.46 \pm 0.13$ | $0.31 \pm 0.13$ | $0.17 \pm 0.13$ |

(a) Action-prediction loss (APL) and intra-cluster trajectory distance (ICTD).

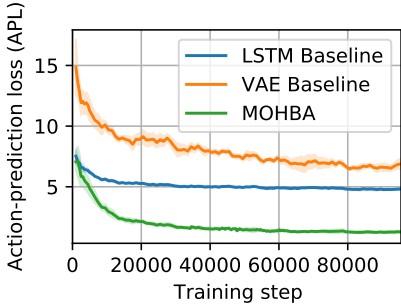

(b) APL convergence throughout training.

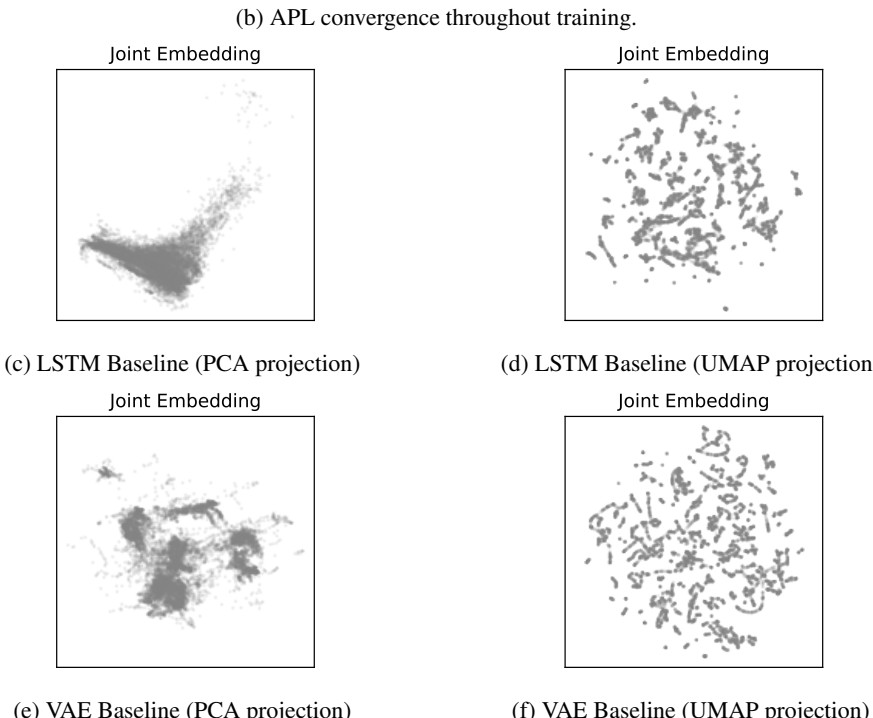

(c) LSTM Baseline (PCA projection)     (d) LSTM Baseline (UMAP projection)

(e) VAE Baseline (PCA projection)      (f) VAE Baseline (UMAP projection)

Figure 8: Baseline comparisons (hill-climbing domain).

result in clusters that overlap more / are 'softer' and slightly less distinguishable (e.g., comparing $z_\omega$ in Fig. 12c versus Fig. 12d); this also aligns well with intuition, as increasing the $\beta$ term prioritizes the KL divergence between the posterior and prior, which deprioritizes disentangling the behaviors for the policy reconstruction term (5).

Overall, these results provide us intuition in terms of the role of these hyperparameters in the behavior clusters learned. At a high level, it appears that the sensitivity of the results to these hyperparameters is fairly low (in the range of values explored in these experiments), thus allowing the high level behaviors discovered to remain reasonably distinctive across the various sweeps.

|  | APL | ICTD | | |
|---|---|---|---|---|
|  |  | K=4 | K=8 | K=16 |
| LSTM | $2.9 \pm 0.1$ | $0.41 \pm 0.15$ | $0.40 \pm 0.14$ | $0.34 \pm 0.15$ |
| VAE | $5.0 \pm 0.3$ | $0.18 \pm 0.11$ | $0.17 \pm 0.11$ | $0.16 \pm 0.10$ |
| MOHBA | $2.4 \pm 0.3$ | $0.18 \pm 0.12$ | $0.17 \pm 0.10$ | $0.16 \pm 0.10$ |

(a) Action-prediction loss (APL) and intra-cluster trajectory distance (ICTD).

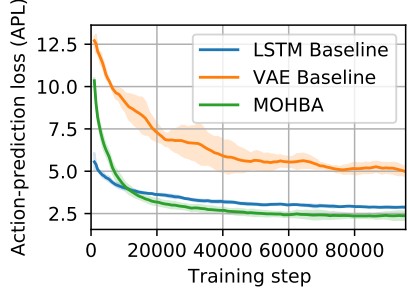

(b) APL convergence throughout training.

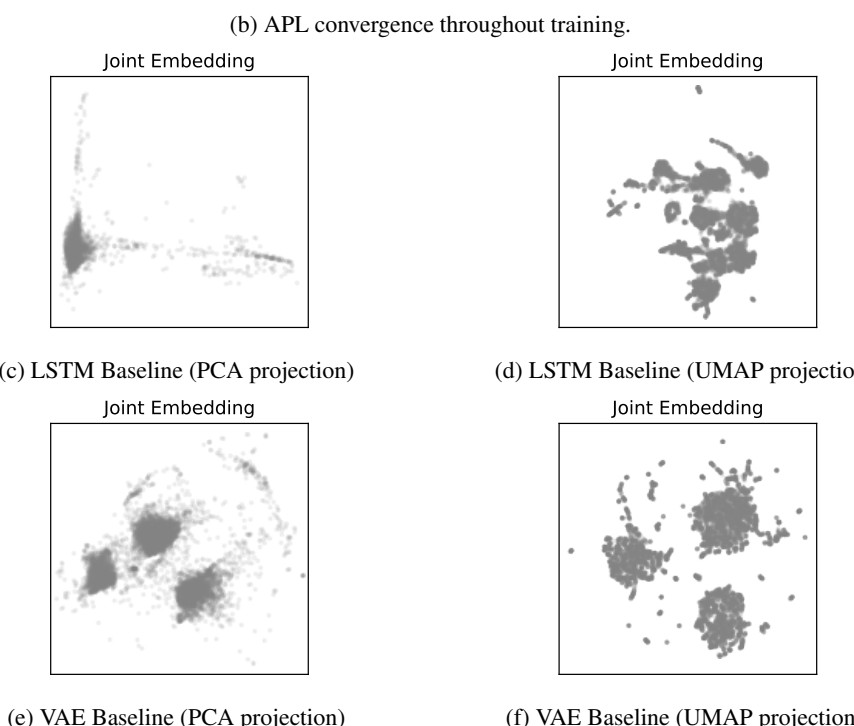

(c) LSTM Baseline (PCA projection)       (d) LSTM Baseline (UMAP projection)

(e) VAE Baseline (PCA projection)       (f) VAE Baseline (UMAP projection)

Figure 9: Baseline comparisons (coordination game).

## A.5   Additional Dataset Details / Reward Distributions

This section provides additional details and statistics regarding the trajectory datasets used for evaluation. Specifically, one of the core properties of datasets typically used for offline RL is the inclusion of a large number of sub-optimal and random trajectories [85]. As noted earlier, the trajectory data for the agents in our experiments is collected from numerous underlying MARL training runs for each domain. To help ensure the collection and analysis of a diverse set of behaviors including sub-optimal ones, we not only collect trajectories at the end of MARL training, but rather from policy checkpoints throughout all of training. All such trajectories, including those stemming from randomly-initialized agent policies at the beginning of training, are shuffled together and used to simultaneously train MOHBA.

Table 7: Trajectory return statistics for each of the datasets analyzed.

| Domain | % of trajectories with total return $< 0.5\times$ (maximum observed return) |
|---|---|
| HalfCheetah (2 agents) | 92.69% |
| AntWalker (4 agents) | 22.47% |
| Hill-climbing | 23.91% |
| Coordination game | 11.83 % |

Here we more closely inspect our datasets to better understand the distribution of agent behaviors. At a high level, we first note that our datasets consist of a significant proportion of sub-optimal trajectories. Specifically, in Table 7, we note the percent of trajectories in each of the considered datasets that attain less than 50% of maximum observed return. Note that in the high-dimensional MultiAgent MuJoCo environments especially, a large proportion of low-reward trajectories exist in the dataset, with the HalfCheetah dataset being a particularly notable one where 92.69% of datasets attain less than 50% of max observed return.

We additionally provide a more detailed overview of trajectory returns in each of Figs. 13a, 14a, 15a and 16a. These figures visualize the distribution of individual agent returns in each of the considered environments (respectively, hill-climbing, coordination game, 2-agent HalfCheetah, and 4-agent AntWalker). Notably, many of the trajectories in the HalfCheetah domain are skewed towards medium and low-return behaviors, with very few high-return trajectories present (Fig. 13a). Interestingly, in the AntWalker dataset we observe a bimodal return distribution (seemingly consisting of very low return and very high return trajectories, as seen in Fig. 14a). Closer inspection of this domain reveals that a large number of training runs result in highly sub-optimal behaviors with very low returns; in Fig. 14b, we visualize the distribution of returns with these low-return trajectories excluded, observing a similar distribution to the HalfCheetah dataset (consisting primarily of medium-return trajectories). Similarly, a large proportion of trajectories with extremely low returns exist in the hill-climbing and coordination game domains (Figs. 15a and 16a); we verified that these low-return trajectories consist primarily of random agent behaviors collected early in training. Overall, this analysis is helpful in determining that the analyzed datasets exhibit a wide range of trajectories (in terms of agent returns), covering random behaviors, sub-optimal behaviors, and high-return behaviors, similar to typical offline RL datasets [85].

As part of this investigation, we also analyze the discovered behavior spaces (both at joint and local agent levels) with respect to the return distributions. Specifically, Figs. 13b, 14c, 15b and 16b visualize the behavior spaces learned by MOHBA, which are then labeled by the ground truth return corresponding to each trajectory. Interestingly, despite MOHBA not using *any* reward or return information in learning agent behavior spaces, clear clusters corresponding to low-return (random/early-training trajectories), medium-return trajectories, and high-return trajectories are automatically discovered in all of the considered domains. For example, in the hill-climbing environment, many of the trajectories corresponding to low returns are clustered in the center of the joint latent space in Fig. 15b; in the HalfCheetah environment, a prominent cluster of high-return trajectories is visible in the top-left and bottom-left local latent spaces of agents 0 and 1, respectively (Fig. 14c); in the AntWalker environment, the lowest-return trajectories are clustered in distinctive regions both at the joint and local level, with a medium-return trajectory set also visible in the latent spaces (Fig. 14c). Overall, these results indicate the richness of the learned latent spaces in terms of not only raw trajectories but also their capacity to cluster trajectories that exhibit similar performance, even without observing the reward function.

### A.6  Visualizing $z_\omega$ and $z_\alpha$

This section summarizes the procedure used to generate the behavior space figures associated with $z_\omega$ and $z_\alpha$. The visualization procedure we use is as follows:

1. We train MOHBA using the offline behavior datasets of interest.

2. Following training of MOHBA, we pass each trajectory $\tau$ through both the joint encoder $q_\phi(z_\omega|\tau)$ and the agent-wise local encoders $q_\phi(z_\alpha^i|\tau)$, sampling a joint latent $z_\omega$ and a set of

local latents $z_\alpha^i$ for all $N$ agents accordingly. Thus, for $K$ such trajectories we obtain a set of latent parameters $\{(z_\omega, z_\alpha^1, \cdots, z_\alpha^N)_k\}_{k=1}^K$.

3. As the latent parameters are high-dimensional in nature, we visualize their 2D projection (e.g., using Principal Component Analysis), thus yielding the behavior space visualizations such as those in Fig. 3a.

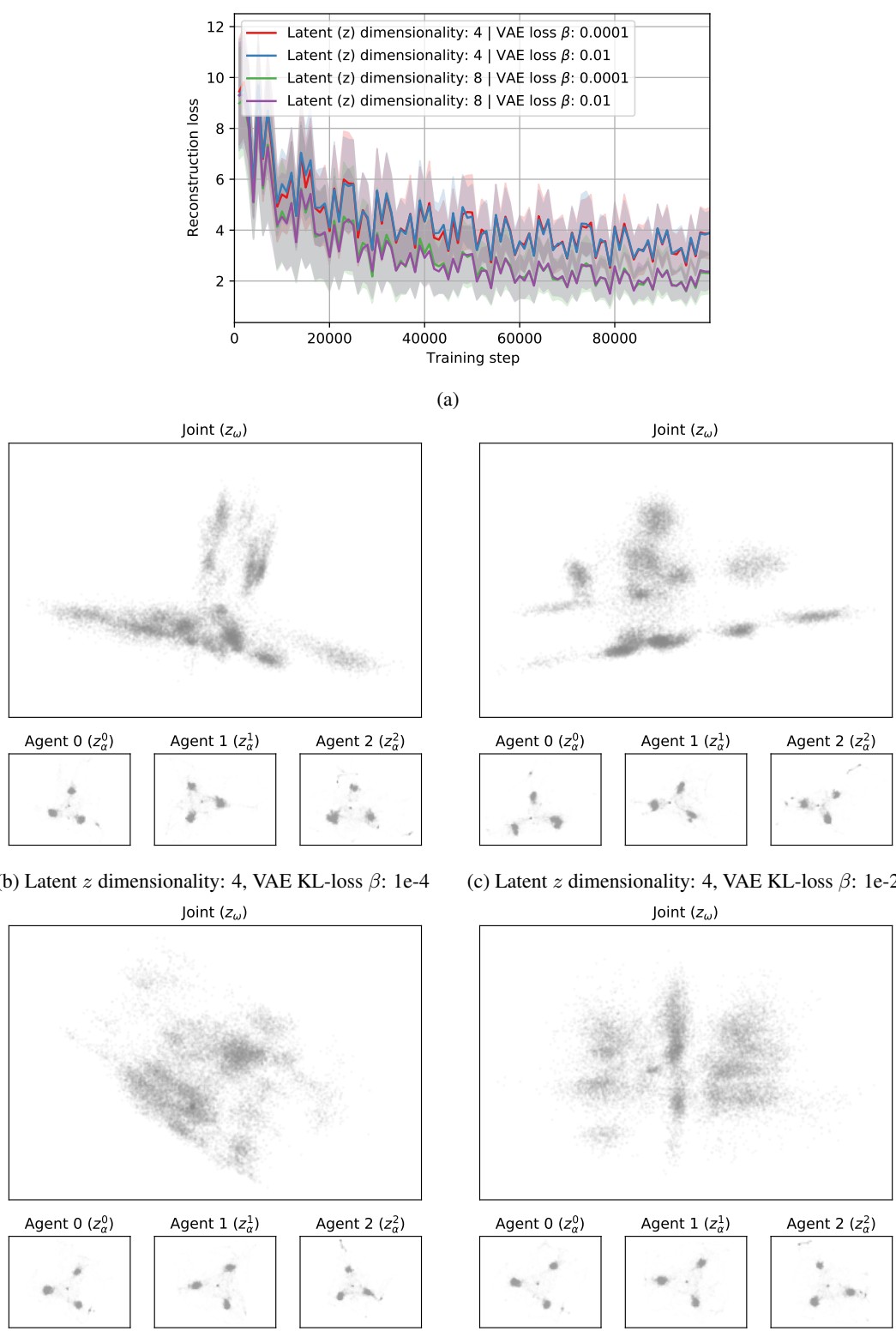

(a)

(b) Latent $z$ dimensionality: 4, VAE KL-loss $\beta$: 1e-4

(c) Latent $z$ dimensionality: 4, VAE KL-loss $\beta$: 1e-2

(d) Latent $z$ dimensionality: 8, VAE KL-loss $\beta$: 1e-4

(e) Latent $z$ dimensionality: 8, VAE KL-loss $\beta$: 1e-2

Figure 10: Ablations over hyperparameters for 3-agent hill climbing environment. (a) visualizes the reconstruction policy loss throughout MOHBA training. (b)-(e) visualize the change in latent space for each of the final MOHBA models learned over these ablations.

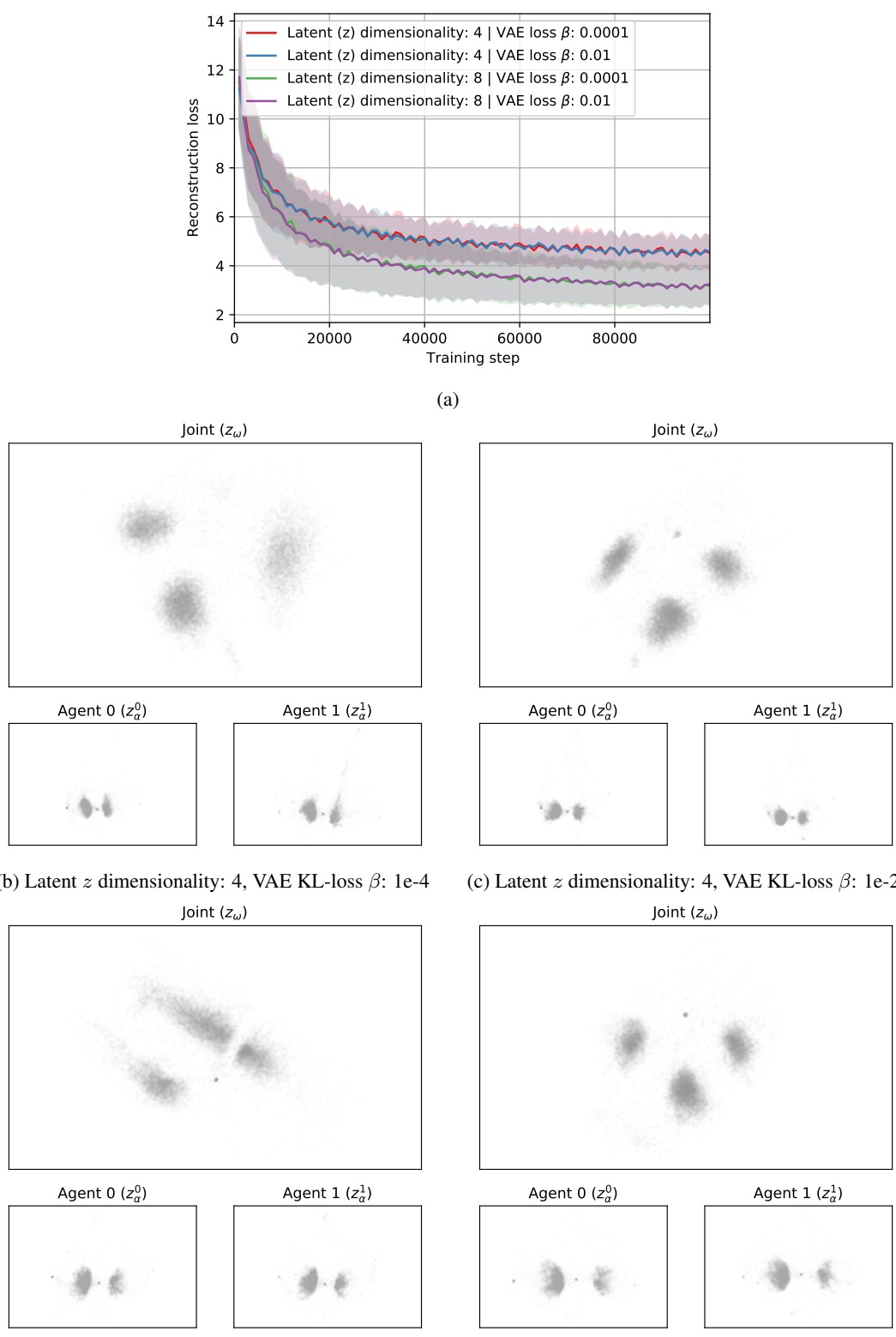

(a)

(b) Latent $z$ dimensionality: 4, VAE KL-loss $\beta$: 1e-4

(c) Latent $z$ dimensionality: 4, VAE KL-loss $\beta$: 1e-2

(d) Latent $z$ dimensionality: 8, VAE KL-loss $\beta$: 1e-4

(e) Latent $z$ dimensionality: 8, VAE KL-loss $\beta$: 1e-2

Figure 11: Ablations over hyperparameters for 2-agent coordination game. (a) visualizes the reconstruction policy loss throughout MOHBA training. (b)-(e) visualize the change in latent space for each of the final MOHBA models learned over these ablations.

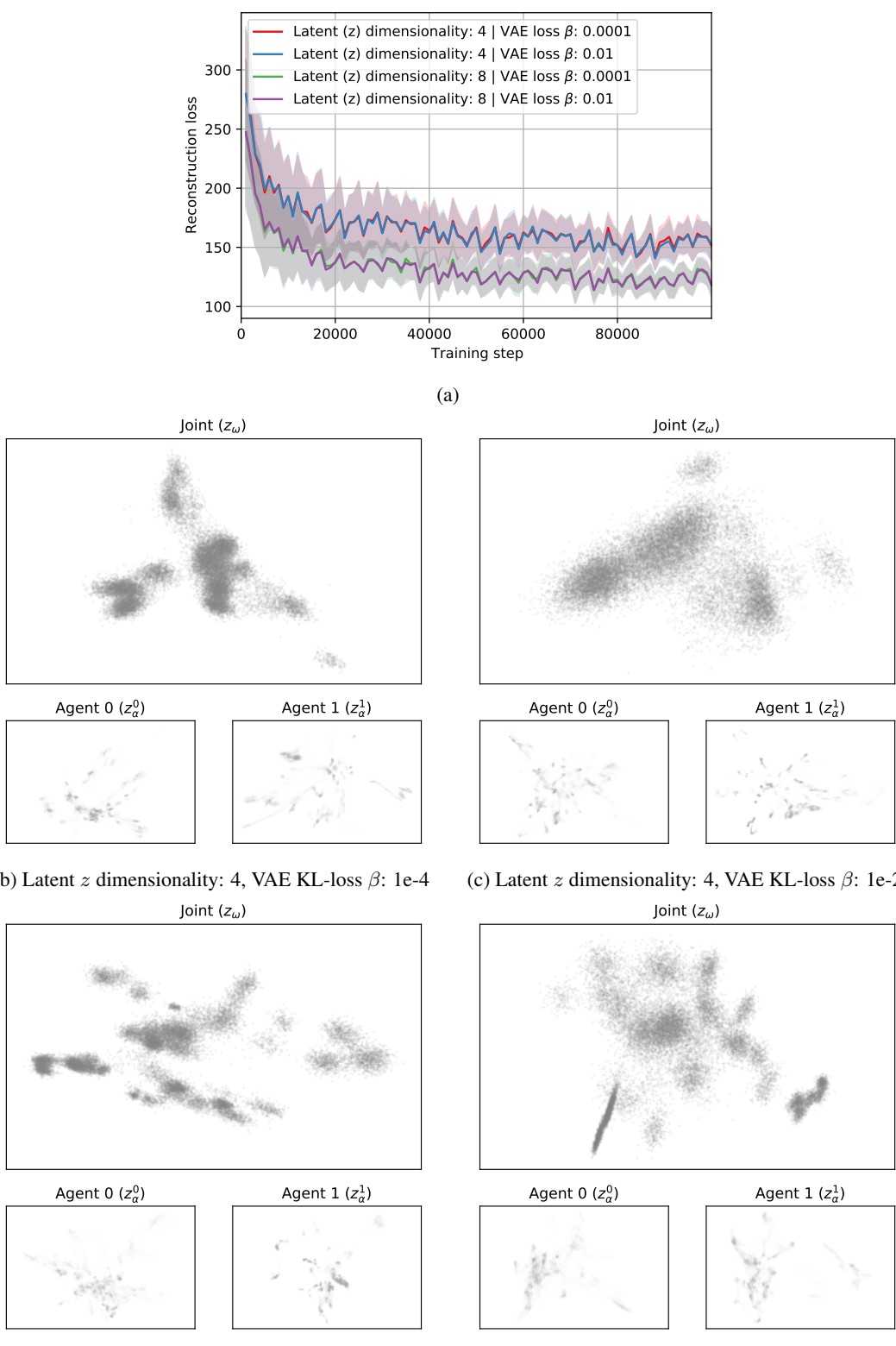

Figure 12: Ablations over hyperparameters for Multiagent MuJoCo HalfCheetah. (a) visualizes the reconstruction policy loss throughout MOHBA training. (b)-(e) visualize the change in latent space for each of the final MOHBA models learned over these ablations. Increasing the dimensionality of $z$ tends to increase the number of joint clusters identified (e.g., (c) versus (e)). Increasing the KL $\beta$ term from $1e-4$ to $1e-2$ tends to result in clusters that overlap more / are 'softer' and slightly less distinguishable (e.g., comparing $z_\omega$ in (c) versus (d)).

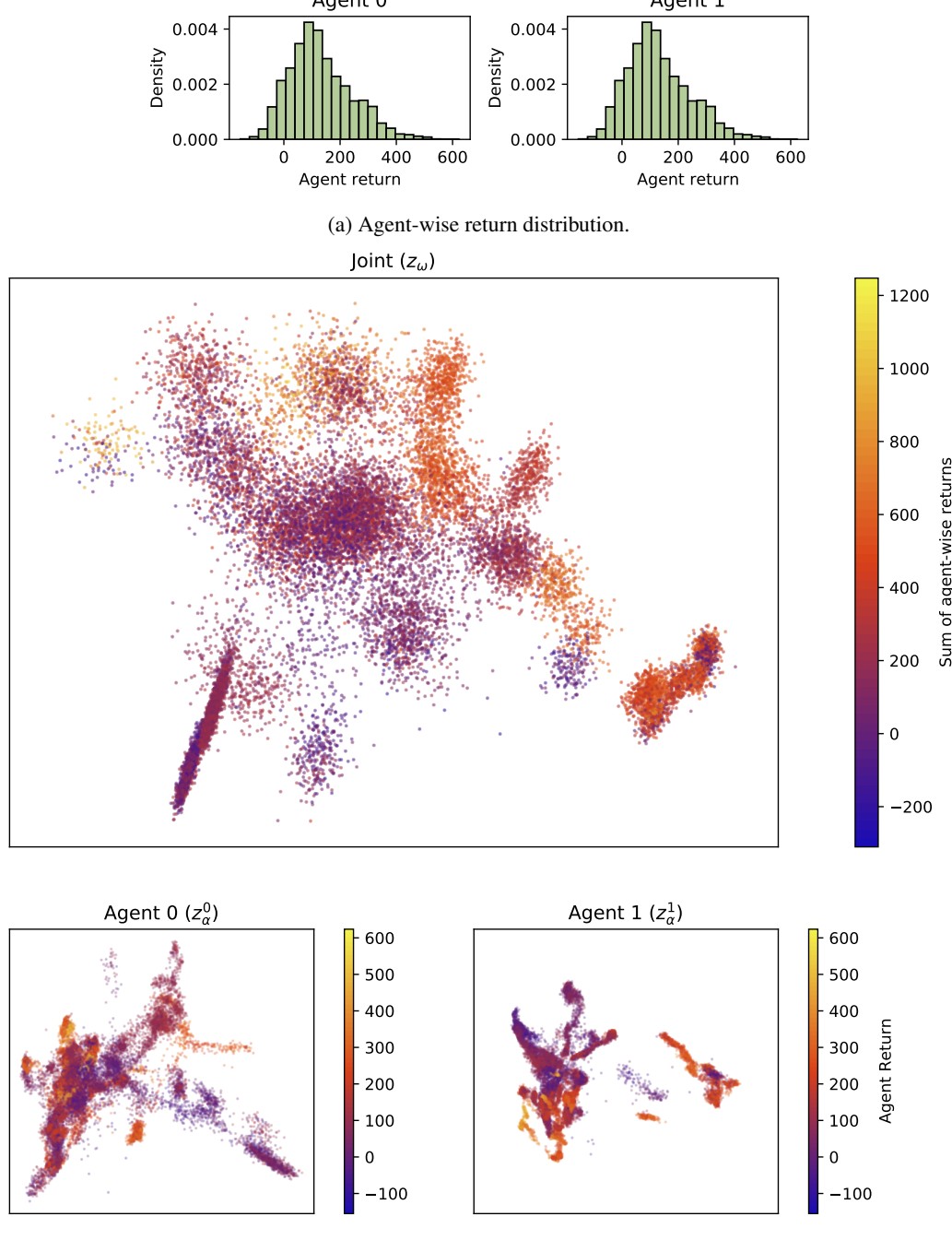

(a) Agent-wise return distribution.

(b) Latent spaces, labeled by trajectory return.

Figure 13: Reward statistics for the Multiagent MuJoCo 2-agent HalfCheetah environment. (a) visualizes the distribution of agent returns in the dataset. (b) visualizes the agent behavior spaces, with trajectories labeled by the return attained.

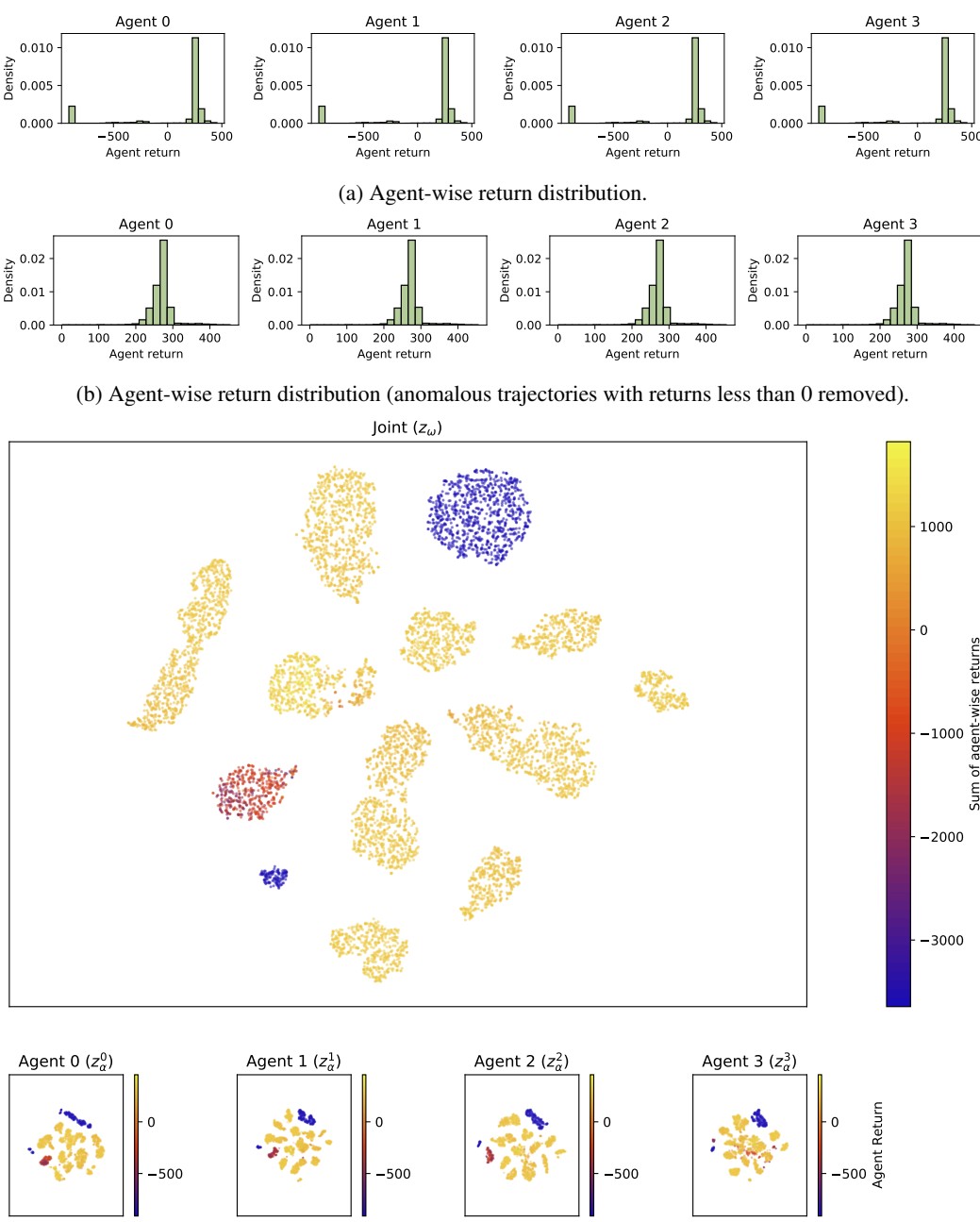

(a) Agent-wise return distribution.

(b) Agent-wise return distribution (anomalous trajectories with returns less than 0 removed).

(c) Latent spaces, labeled by trajectory return.

Figure 14: Reward statistics for the Multiagent MuJoCo 4-agent AntWalker environment. (a) visualizes the distribution of agent returns in the dataset. Interestingly, we observe a bimodal return distribution (seemingly consisting of low return and higher return trajectories. Closer inspection reveals that a large number of MARL training runs in this domain result in highly sub-optimal behaviors with very low returns; in (b), we visualize the distribution of returns with these low-return trajectories excluded, which better illustrates the distribution of medium-high reward trajectories in this dataset. (c) visualizes the agent behavior spaces, with trajectories labeled by the return attained.

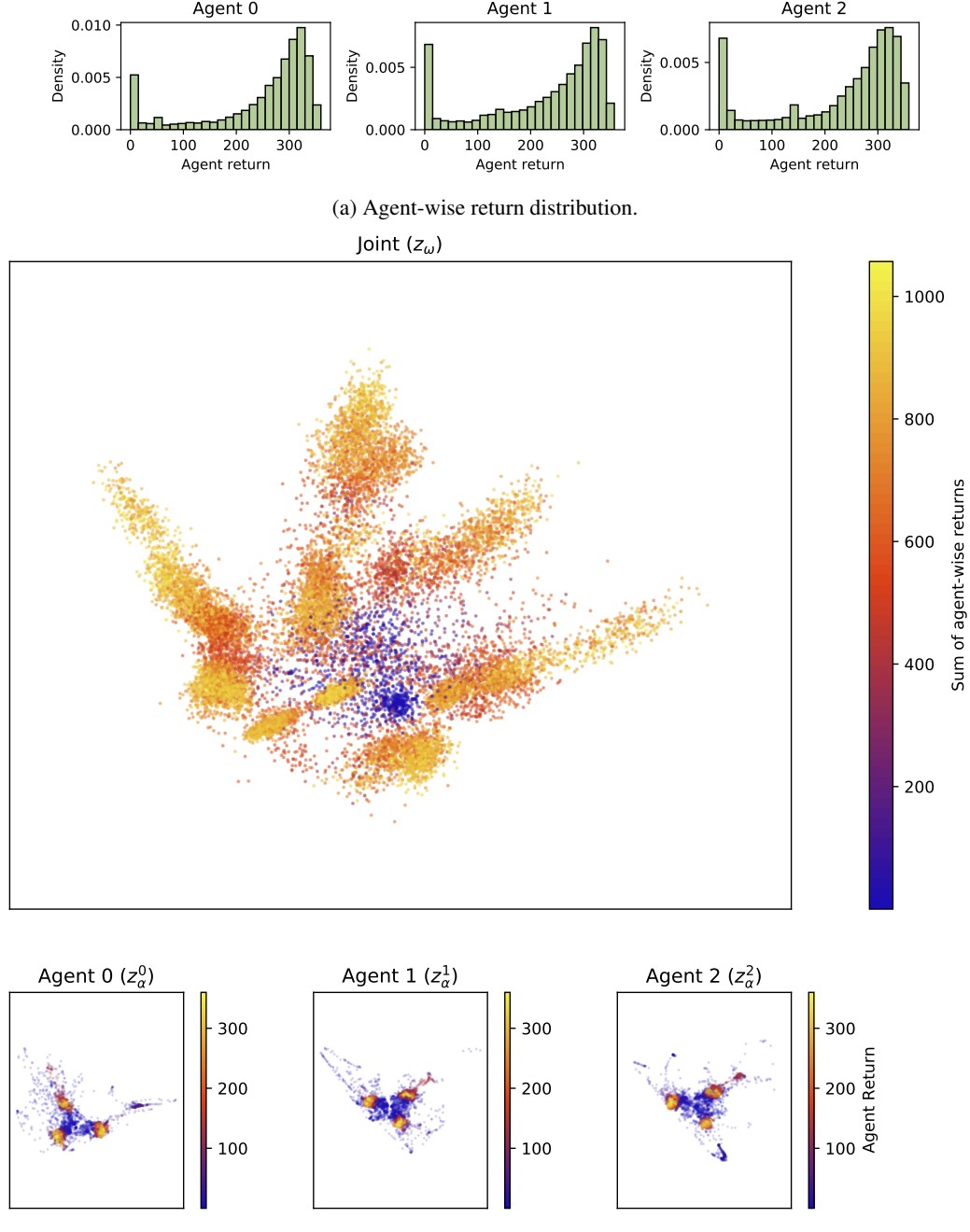

(a) Agent-wise return distribution.

(b) Latent spaces, labeled by trajectory return.

Figure 15: Reward statistics for the 3-agent hill climbing environment. (a) visualizes the distribution of agent returns in the dataset. (b) visualizes the agent behavior spaces, with trajectories labeled by the return attained.

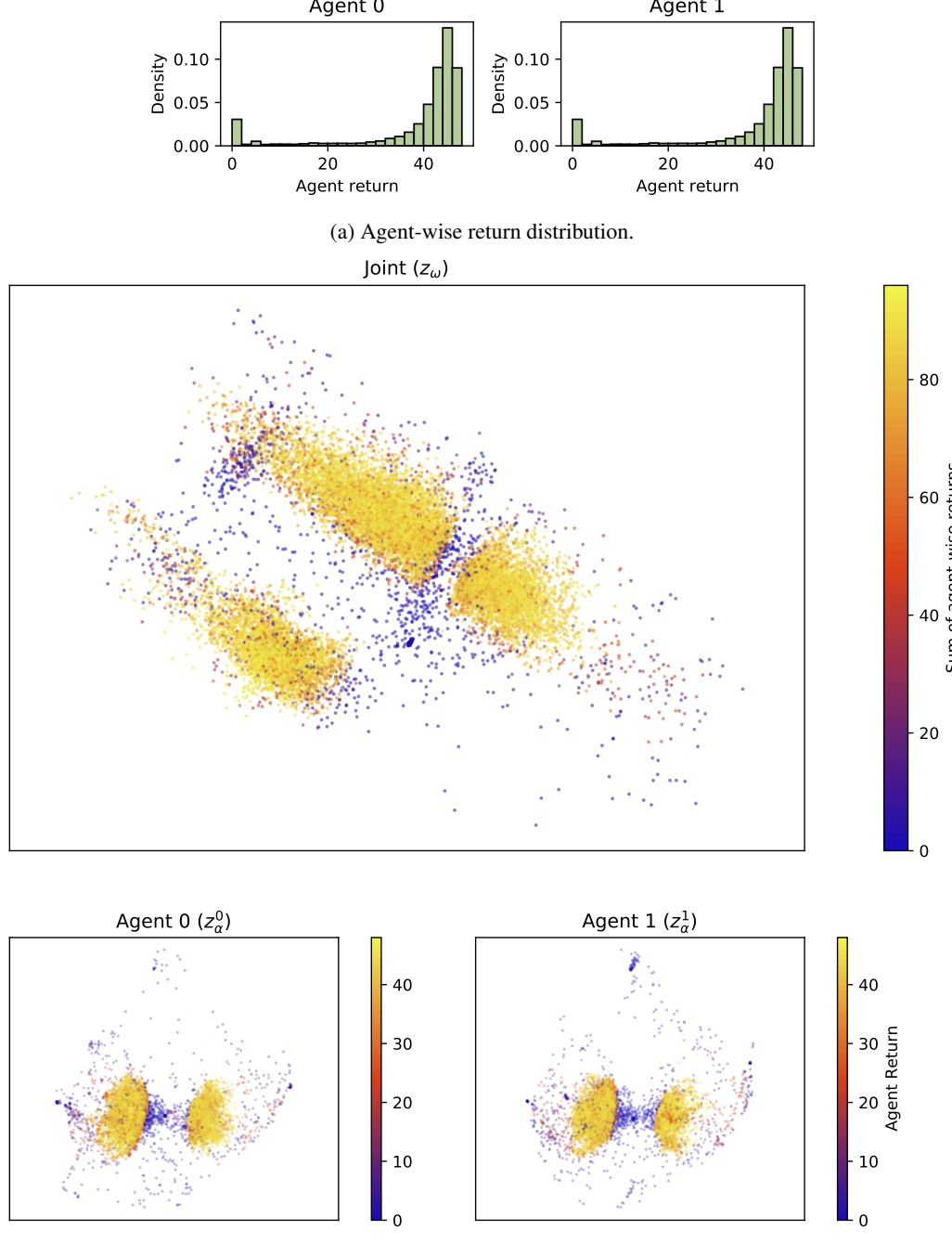

(a) Agent-wise return distribution.

(b) Latent spaces, labeled by trajectory return.

Figure 16: Reward statistics for the 2-agent coordination environment. (a) visualizes the distribution of agent returns in the dataset. (b) visualizes the agent behavior spaces, with trajectories labeled by the return attained.

## A.7 Model high-level code

Listing 1 provides the high-level code of the MOHBA model. Note that the components of our model are standard network modules (e.g., MLPs, bidirectional LSTMs, etc.), which are indicated below and can be implemented using any desired neural network library.

```python
"""Multiagent Offline Hierarchical Behavior Analyzer (MOHBA)."""

from typing import Dict, Tuple, Union
import distrax
from flax import linen as nn
import jax
import jax.numpy as jnp

Array = Union[np.ndarray, jnp.ndarray]

class MOHBA(nn.Module):
  config: Dict
  num_agents: int
  num_actions_per_agent: int

  def setup(self):
    """Setup model components."""
    # Joint models
    self.joint_prior = GaussianMixture(self.config.joint_prior)
    self.joint_encoder = GaussianMixtureBidirLSTM(self.config.joint_encoder)

    # Local models (shared parameters across agents)
    self.local_prior = MLP(**self.config.local_prior)
    self.local_encoder = BidirLSTM(**self.config.local_encoder)
    self.local_policy = MLP(**self.config.policy)

  def __call__(self, rng: Array, states: Array, actions: Array):
    """Returns model outputs and KL divergences for loss computation.

    Args:
      rng: jax rng state.
      states: joint states, shape [B, T, N, S].
      actions: joint actions, shape [B, T, N, A].
    """
    B, T = states.shape[:2]
    agents_onehot = get_agents_onehot(
        front_dims=(B, T), num_agents=self.num_agents)

    # Append agent-onehots to tau.
    tau = jnp.concatenate((states, actions), axis=-1)
    tau_with_onehots = jnp.concatenate((tau, agents_onehot), axis=-1)

    ### Joint prior (z_omegas) ###
    dist_joint_prior = self.joint_prior(B)
    rng, z_omegas_rng = jax.random.split(rng)

    # self.joint_encoder internally reshapes tau to consume joint agent information
    dist_joint_encoder = self.joint_encoder(tau)
    z_omegas = dist_joint_encoder.sample(seed=z_omegas_rng)
    # Append agent one-hots to z_omegas
    z_omegas_with_onehots = utils.expand_and_concat_agent_ids(
        z_omegas, self.num_agents)  # [B, L] to [B, N, L+N]

    ### Local priors (z_alphas) ###
    # vmap over dim 1 of z_omegas_with_onehots (shape [B, N, L+N])
    dist_params = jax.vmap(
        self.local_prior.dist_params, in_axes=1, out_axes=1)(z_omegas_with_onehots)
    dist_local_prior = distrax.MultivariateNormalDiag(
        loc=dist_params[0], scale_diag=jnp.exp(dist_params[1]))

    rng, z_alphas_rng = jax.random.split(rng)
    dist_params = jax.vmap(
        self.local_encoder.dist_params, in_axes=2, out_axes=1)(tau_with_onehots)
    dist_local_encoder = distrax.MultivariateNormalDiag(
        loc=dist_params[0], scale_diag=jnp.exp(dist_params[1]))
```

```
67      ### Agent policies ###
68      z_alphas = dist_local_encoder.sample(seed=z_alphas_rng)
69      z_alphas_policy = jnp.repeat(z_alphas[:, jnp.newaxis], T, axis=1)
70      # Parameter-shared policies are run over each agent's own states
71      pred_actions = jax.vmap(self.local_policy, in_axes=2, out_axes=2)(
72          states, agents_onehot, z_alphas_policy)
73
74      # KL divergence for joint latents.
75      rng, kl_rng = jax.random.split(rng)
76      kl_joint_latents = distrax.kl_divergence(
77          distribution_a=dist_joint_encoder, distribution_b=dist_joint_prior)
78
79      # KL divergence for local latents.
80      rng, kl_rng = jax.random.split(rng)
81      kl_local_latents = distrax.kl_divergence(
82          distribution_a=dist_local_encoder, distribution_b=dist_local_prior)
83      return pred_actions, kl_joint_latents, kl_local_latents
84
85
86  def get_agents_onehot(front_dims: Tuple[int, ...], num_agents: int):
87    """Returns one-hot representation of agents for batched data.
88
89    E.g., get_agents_onehot(front_dims=(2,3), num_agents=5) returns an
90    array of shape [2,3,5,5], where the last two dims specify one-hots for each of
91    the 5 agents, repeated over the first two `front_dims`.
92
93    Args:
94      front_dims: tuple specifying the sizes of the first K dims.
95      num_agents: the number of agents.
96    """
97    agents_onehot = np.arange(num_agents)
98    agents_onehot = jax.nn.one_hot(agents_onehot, num_agents, axis=-1)
99    agents_onehot = jnp.broadcast_to(
100     agents_onehot, front_dims+(num_agents, num_agents))
101   return agents_onehot
102
103
104 def expand_and_concat_agent_ids(x: Array, num_agents: int):
105   """Expand input x dims and concatenate agent ID one-hots to its features.
106
107   Specifically, given input x of shape [B, features], adds agents dim,
108   then concatenates agent ID one-hots to the features dim, resulting in output
109   of shape [B, num_agents, features+num_agents].
110
111   Args:
112     x: input, shape [B, features].
113     num_agents: the number of agents.
114
115   Returns:
116     x: output, shape [B, num_agents, features+num_agents].
117   """
118   # Create [B, num_agents, num_agents] agent IDs one-hot tensor
119   agents_onehot = get_agents_onehot(
120     front_dims=(x.shape[0],), num_agents=num_agents)
121
122   # Expand x from [B, features] to [B, num_agents, features]
123   x = jnp.expand_dims(x, axis=1)
124   x = jnp.repeat(x, num_agents, axis=1)
125
126   # Concat x and onehots
127   return jnp.concatenate((x, agents_onehot), axis=-1)
```

Listing 1: MOHBA model high-level code.