# OpenReview forum: "Beyond Rewards: a Hierarchical Perspective on Offline Multiagent Behavioral Analysis"
_NeurIPS.cc/2022/Conference — NeurIPS 2022 Accept_

### Official Review · Reviewer_do8M · 2022-07-06

**Rating:** 7
**Confidence:** 4
**Soundness:** 4 excellent
**Presentation:** 4 excellent
**Contribution:** 3 good

**Summary:**

This paper studies an interesting question: how to model different behaviors from an offline multi-agent RL dataset. The authors propose a hierarchical modeling approach and present empirical experiments over a 2-D particle world navigation scenario and a MuJoCo HalfCheetah control task.


++++++++ post-rebuttal +++++++++
I think the updated draft is clearly a much stronger one as a solid work. I have updated my score accordingly.

**Questions:**

I'm curious about answers to the following questions:

1. How does the proposed method perform if the dataset includes a large portion of sub-optimal and random trajectories?
2. How does the proposed method perform on more complex domains with more agents?

**Strengths And Weaknesses:**

### Strengths:
I personally like the topic of this paper and I strongly agree that the analysis of RL applications should be beyond rewards, i.e., considering the behaviors of the learned strategies. The use of a hierarchical generative model is intuitive (although it isn't surprising). I do feel this draft has the potential to become a much stronger paper if the authors can step further in this research direction.
Regarding the writing, the method and motivation part is clearly written but I don't think many of the arguments from this paper are well justified.

### Weakness:
The current content looks a bit preliminary. There are two major issues: (1) the technical content is not well justified. (2) the experiments are a bit too toy for an analysis paper. Please see my concerns below.

1. **Regarding the technical content:** The use of the generative model in modeling agent behaviors isn't new at all. Lots of techniques, such as [GAN-style](https://arxiv.org/abs/1802.06070), [VAE](https://arxiv.org/abs/2010.01523), and [diffusion model](https://diffusion-planning.github.io/), have been widely adopted. The authors choose a VAE-style generative modeling approach, which is technically standard. The interesting part appears to be the use of two latent variables, i.e., one for joint behavior and the other for agents. But I don't really get why this hierarchical design is necessary. At least for the motivating example (the hill climb game), a flattened VAE model is clearly sufficient. Even in the experiment part (Table 2), I cannot see any critical advantages ------ yes, it is true that the bi-level model has a higher prediction accuracy but why do we really care about prediction accuracy? From the motivation of the paper, the paper focuses on behavior analysis and aims to discover different behavior modes, which are well captured by the ICTD metric. I couldn't see a big difference between the two methods under the ICTD metric. Moreover, at least from the current content of the paper, I think both the half cheetah and the MPE motivating example can be perfectly analyzed by the flattened model for the purpose of discovering strategy modes. I do agree that the bi-level model discovers some agent-wise behavior modes. But so what? If we are really interested in agent-wise behavior analysis, we can simply run another flattened VAE model for each agent over its own trajectories and that's all set. To sum up, the technical content of the paper looks so arbitrary. Some further analysis and research would be required for a solid work.
2. **Regarding the experiment section:** First of all, the experiment section is way too toy for an analysis paper. It is okay to take the hill-climb environment as a motivating example. But I would expect much more complex domains. The MoJoCo control domain is a good example. But why only the simplest scenario with just two agents is considered? Why not consider more scenarios with much more agents? The authors talk about the hide-and-seek project in the introduction section repeatedly. So why not run the model over the hide-and-seek project? The learned policies are released. You don't even need to train the policies. Moreover, the author claimed in the introduction section that the proposed method is reward-agnostic and is generic to offline RL settings. However, the dataset construction doesn't really follow the offline RL convention. The dataset is simply constructed by running MARL training repeatedly. Note that the MARL algorithms are optimizing the environment reward, which implies all the behavior data are leaning towards high rewards. What if some sub-optimal or even random trajectories are included in the dataset? Can the model still work? We would like to point out that including sub-optimal and random trajectories is a common practice in [offline RL literature](https://arxiv.org/abs/2004.07219).

In general, I would suggest the authors re-think the emphasis of the paper. What is really the main contribution? Is it the hierarchical modeling technique? Or the behavior analysis problem itself? If the paper focuses on the former one (bi-level modeling), the motivating example should make people understand why a bi-level model is necessary instead of a flattened model. The experiments should also focus more on the bi-level v.s. flatten part. If the authors would like to focus more on a new problem, then we would expect much more experiments and analysis, hopfully including novel insights or findings beyond cheatah skills (there are so many papers showing different emergent motion skills over MuJoCo domains).


### Minor Issues
1. **A missing line of research**: I would suggest the author also take into account the works on RL diversity, which focuses on learning a collection of policies covering different behavior modes. These works typically use a discriminative approach to model behavior modes, i.e., defining a distance metric or a heuristic density, rather than a generative model. I would like to point out that a distance metric can be perfectly used for clustering although these methods simply optimize the policies to ensure diversity (remark: optimizing diverse policies is a much harder problem than simply clustering over a fixed dataset.) Representative metrics include [trajectoriy diversity](https://proceedings.mlr.press/v139/lupu21a.html), [DvD](https://github.com/jparkerholder/DvD_ES), [cross-entropy](https://openreview.net/forum?id=hcQHRHKfN_), and [quality diversity](https://www.frontiersin.org/articles/10.3389/frobt.2016.00040/full).. I would suggest the authors include this line of research and, if possible, discuss why a generative modeling approach is preferred here? E.g., why cannot we just design a quality diversity function and run a t-SNE analysis or just run a kernel-version of GMM?
2. **Experiments on _``user-desired characteristics''_**. I don't really get the motivation of this part (line 239). Why does "user-desired" necessarily mean "high rewards"? Also, as I mentioned above, the dataset is constructed by running MARL algorithms, so many of the trajectories are indeed having high rewards. I wouldn't be surprised that high-reward concepts can be learned. But what if more random data are introduced? What if most data are random? In addition, I couldn't understand why a train-validation split makes any sense here. Why do we really care about testing accuracy? The authors state that they are following [13]. However, I would point out that [13] is in the setting of supervised learning, which is different from RL. Why could all the evaluation settings of SL be simply generalized to RL? It is natural for supervised learning methods to focus on loss function/prediction accuracy. But why do we care about the loss in RL? Note that even the title of the paper claims "beyond rewards". Should we measure something beyond prediction accuracy? I would suggest the author think a bit further to present more convincing experiments.

---

> ### Author Response · Authors · 2022-08-02
> **Author Response to Reviewer do8M (Part 1)**
>
> We thank the reviewer for the positive feedback regarding the importance of the multiagent behavioral analysis problem setting, and clarity of the method and motivation.  We especially also appreciate and thank you for the detailed constructive feedback, which we have used to revise the paper accordingly. We believe these changes have resulted in a significantly stronger version of the paper.
>
> We have substantially improved the paper as a result of your feedback including substantial new experiments (such as on OpenAI hide and seek), and updated the paper PDF (which now includes both the main paper + Supplementary Information) accordingly. Our key additions are summarized as follows:
>
> - Added new experiments for 4-agent OpenAI hide and seek policy checkpoints, where our method learns to distinguish ‘running and chasing’, ‘fort building’, ‘ramp use’, and ‘ramp defense’ behaviors [Supplementary Information A.3.1]
> - Added new experiments for 4-agent MultiAgent MuJoCo AntWalker [Supplementary Information A.3.1]
> - Conducted new experiments analyzing effects of key hyperparameter on results [Supplementary Information A.4]
> - Investigated the proportions of random, sub-optimal, and expert trajectories in our datasets, and conducted additional reward-based experiments [Supplementary Information A.5]
> - Adding all suggested related works to discussions
> - Added numerous clarifications and added contexts throughout the text.
>
> Please note that due to the page limit still applying in the rebuttal period and due to limited time, we have included expanded experiments in the Supplementary Information, but intend to move them to the main text and move/shorten previously-existing sections to prioritize these latest results (especially on OpenAI hide and seek). All figure & table # references below refer to this updated revision.
>
> Please find our point-by-point responses below.

---

> > ### Author Response · Authors · 2022-08-02
> > **Author Response to Reviewer do8M (Part 2)**
> >
> > —-------------------------—-------------------------—-----------
> >
> > 1) Technical content and main contribution of the paper and the need for further analysis.
> >
> > Response: We thank the reviewer for this constructive feedback regarding the contribution and emphasis of the paper. Indeed, the primary focus of the paper is to emphasize the multiagent behavioral modeling problem. While previous works in MARL (and general multiagent systems) have conducted various forms of ad-hoc evaluation of emergence of interesting agent behaviors, our view is that there still remains a strong focus on performance-driven techniques, whereas post-hoc analysis is not as prominently focused upon. We believe that the publishing of papers such as ours can help to balance the community’s focus towards better understanding of not just ‘what’ multiagent systems are capable of doing, but ‘how’ they are capable of doing so.
> >
> > As noted by the reviewer, we leverage the substantial work done in the past on generative modeling and apply this technique to this new problem regime of interest. As such, the proposed bi-level hierarchical approach is meant to provide a means of conducting analysis in this new problem regime, using a method that performs well across a variety of domains (including OpenAI hide and seek in our latest experiments), makes few assumptions about the dataset (i.e., no access to the environment or underlying agent policies/hidden states) and associated level of annotations provided (i.e., using only raw observational data), and provides an explicit link between joint and local agent behaviors.
> >
> > While the ICTD metric in our baseline experiments provides one measure of comparing the flat-VAE approach to ours in the simple 2-dimensional domains, designing such metrics for domains with significantly more intricate state-spaces (e.g., combinations of various agent positions, rotations, sensory measurements, etc.) is non-trivial. The core difference between repeated runs of a flat-VAE approach and our approach is that ours models the relationship between joint-level and individual agent-level behaviors in a single, unified model, which is helpful for understanding the progression of coordinated team behaviors throughout training. Our hypothesis is that such approaches will not only allow us to identify potentially misbehaving agents during MARL training (as in Figure 4d of our coordination game results), but also reasoning between coordinated/joint behavior differences (such as ‘fort building’ versus ‘ramp defense’ for ‘hider’ agents in our OpenAI hide and seek experiments).
> >
> > Overall, we completely agree with the reviewer’s point regarding clarifying the focus of the paper, which is on the analysis of this new problem regime. As such, we took your suggestion of pushing this latter direction of investigation further by focusing our efforts on wider experiments and analysis during the rebuttal phase, and added a wide variety of new experiments and larger-scale domains (detailed in the next rebuttal point). We especially wish to highlight our new experiments on the OpenAI hide and seek domain, which illustrate the effectiveness of our proposed approach even in discovering complex and nuanced behaviors in large-scale and complex interactive environments (similar to those labeled by human experts, e.g., ‘fort building’ and ‘running and chasing’). We hope our revision and surrounding discussions have helped to address your concerns and questions.
> >
> > —-------------------------—-------------------------—-----------
> >
> > 2) “The experiments are a bit too toy for an analysis paper” and “How does the proposed method perform on more complex domains with more agents?”
> >
> > Response: As a result of your and other reviewers’ feedback, during the rebuttal period, we substantially expanded the experiments section, including the addition of:
> >
> > a) results for two large-scale new domains, including one with policies trained by external authors (OpenAI hide and seek, visualized by the original authors here: https://www.youtube.com/watch?v=kopoLzvh5jY) and an expansion of the Multiagent MuJoCo results in the larger 4-agent AntWalker domain [revised Supplementary Information A.3.1]
> >
> > b) further ablative analysis of our existing datasets (including effects of hyperparameters on learned latent spaces) [revised Supplementary Information A.4]
> >
> > c) analysis of our previous latent spaces from a return-centric perspective, which reveals that MOHBA (our algorithm) learns to cluster trajectories with similar agent returns despite not having access to reward information during training [revised Supplementary Information A.5]

---

> > > ### Author Response · Authors · 2022-08-02
> > > **Author Response to Reviewer do8M (Part 3)**
> > >
> > > Specifically:
> > >
> > > a) Per your suggestion, during the rebuttal phase we analyzed policy checkpoints from the OpenAI hide-and-seek environment (https://github.com/openai/multi-agent-emergence-environments). We summarize results in Section A.3.1 of the Supplementary Information, included below for convenience:
> > >
> > > """
> > > In the OpenAI hide-and-seek environment, two teams of two agents (hiders and seekers) compete against one another in a rich, high-dimensional environment with various dynamic objects (boxes and ramps) that the agents can interact with. OpenAI has open-sourced policy check-points manually annotated by human experts as exhibiting various distinctive multiagent behaviors at key stages of training.  We consider four distinctive policy checkpoints that each correspond to the following human-annotated labels:
> > > A) ‘running and chasing’, B) ‘fort building’, C) ‘ramp use’, and D) ‘ramp defense’.
> > >
> > > The state-space for each of the agents at each timestep of the trajectory is 100-dimensional, consisting of its own state (position, rotation, and velocity information), states of the other 3 agents, and the states of 3 boxes and one ramp in the environment (position, velocity, and box size information); each agent’s action space consists of a 3-dimensional force vector, and a ‘glue’ and ‘lock’ action for interacting with other objects. Overall, this yields a state-action space with dimensionality (4 agents x 105 state-actions), significantly larger than the (2 agents x 9 state-actions) of the HalfCheetah MuJoCo environment. We collect 100 trajectories per policy checkpoint, with each trajectory being 200 decision-steps long. Note that these trajectories have a wide distribution of behaviors, as agent and object initializations are also random in each episode.
> > >
> > > We then mix the trajectories collected from all of the above policy checkpoints together, then train MOHBA on this shuffled dataset. As usual, note that we provide no reward or return information to MOHBA. Figure 7 visualizes the behavior spaces discovered by MOHBA in the hide-and-seek domain. Agents 0 and 1 in this figure correspond to ‘hiders’, whereas agents 2 and 3 are ‘seekers’. We label each of the trajectories in this figure with the human-annotated behavior labels provided by OpenAI.
> > >
> > > In Fig. 7, we observe the presence of clear behavior clusters that correspond well to the expert policy checkpoint labels, both at the joint and local agent levels. Note that this is despite all the policy checkpoint data being shuffled in the dataset being used to train MOHBA.
> > >
> > > Interestingly, for the seekers (agents 2 and3), policy A (‘running and chasing’) is highly distinctive and well-separated from the other behaviors. Interestingly, despite the order of emergent behaviors in the original hide-and-seek MARL training being A→B→C→D, policies B (‘fort building’) and D (‘ramp defense’) appear to be behaviorally slightly closer to one another than C (‘ramp use’) and D, namely in the joint space and also in that of the seeker (agent 2 and 3) latent spaces. One hypothesis is that this could perhaps be due to both the ‘fort building’ and ‘ramp defense’ policies being associated with situations where the seekers cannot easily find the hiders, due to the hiders using obstacles to block entrances (B) and moving ramps to prevent their effective use (D).
> > >
> > > Overall, these experiments help to validate MOHBA’s learned latent spaces using policy labels manually annotated by human experts, and also highlight the applicability of our algorithm in domains with large state spaces.
> > > """
> > >
> > > We additionally expanded the MuJoCo results to the AntWalker domain in Sec. A.3.1 of the Supplementary Information. Here, 4 agents each control a distinct ant leg to coordinate movement towards the +x-direction (Fig. 6a). To collect data here, we conduct a wider MARL parameter sweep using both the TD3 and SAC algorithms, with varying training batch sizes and learning rates to gather a wide dataset of behaviors. We subsequently train MOHBA on this data, visualizing the learned behavior clusters in Fig. 6b. We observe several behavior clusters of interest; notably, a large joint cluster exists in the top-right region of the joint and local returns, which, upon inspection of underlying trajectory videos, corresponds to cases where agents attain extremely low return. Similarly, there exists a smaller cluster in the lower-left region of the joint returns that also attains extremely low performance. The remaining clusters correspond to the agents displaying various ant-poses, andmoving only incrementally. One exceptional cluster also exists in the left region of the joint behavior space, which attains medium-level return (points that are primarily red in color, seen in the bottom left region of Figure 14’s joint return plots). On closer inspection, the AntWalker behavior in this cluster corresponds to one of the agents learning a reasonably good walking gait, while the remaining three agents remain stationary.

---

> > > > ### Author Response · Authors · 2022-08-02
> > > > **Author Response to Reviewer do8M (Part 4)**
> > > >
> > > > b) Next, we also now conduct ablative analysis to understand the effects of key hyperparameters (the latent dimensionality size, and the KL loss $\beta$ term) on the learned latent spaces, and include this in Supplementary Materials Section A.4. Specifically, we compare both the policy reconstruction loss (Eq. 5) across ablations to attain a quantitative comparison of the quality of the latent encodings, and additionally compare the structure of the learned latent spaces themselves as a function of these parameters for a qualitative understanding. At a high level, we find that the sensitivity of the results to these hyperparameters is fairly low (in the range of values explored in these experiments), thus allowing the high level behaviors discovered to remain reasonably distinctive across the various sweeps, as seen in Figures 10 to 12. Please refer to Supplementary A.4 for figures and significant additional details.
> > > >
> > > > c) We additionally investigate the structure of the latent space from a reward-centric perspective, which reveals several novel insights that we summarize in Supplementary Information Section A.5. During the rebuttal period, we conducted additional analysis of MOHBA’s latent spaces, and discovered that MOHBA learns clear clusters corresponding to low-return (random/early-training) trajectories, medium-return trajectories, and expert return trajectories across all of the domains we considered; this is despite MOHBA never observing any agent reward information during training.
> > > > These results help indicate the richness of the learned latent spaces in terms of not only raw trajectories but also their capacity to cluster trajectories that exhibit similar performance, even without observing the reward function.
> > > >
> > > > Overall, we believe these substantially-expanded results help to further validate our general methodology, quality of the datasets analyzed, and applicability to wider settings including policies collected from external open-sourced codebases.

---

> > > > > ### Author Response · Authors · 2022-08-02
> > > > > **Author Response to Reviewer do8M (Part 5)**
> > > > >
> > > > > 3) “How does the proposed method perform if the dataset includes a large portion of sub-optimal and random trajectories?” and “the dataset construction doesn't really follow the offline RL convention. The dataset is simply constructed by running MARL training repeatedly… which implies all the behavior data are leaning towards high rewards”
> > > > >
> > > > > Response: We thank the reviewer for bringing up this point. We believe there is a misunderstanding here potentially stemming from the fact that we could have detailed the data generation process more carefully. The trajectory data for the agents in our experiments is collected from numerous underlying MARL training runs for each domain. To help ensure the collection and analysis of a diverse set of behaviors including sub-optimal ones, we not only collect trajectories at the end of MARL training, but rather from policy checkpoints throughout all of training. All such trajectories, including those stemming from randomly-initialized agent policies at the beginning of training, are shuffled together and used to simultaneously train MOHBA.
> > > > >
> > > > > We agree with the reviewer’s feedback, though, that it would be worthwhile and valuable to investigate the reward distributions of the generated datasets more closely. We investigated this across all considered domains during the rebuttal phase, summarizing our results in Supplementary Information Section A.5, and also below for convenience:
> > > > >
> > > > > At a high level, we first note that our datasets consist of a significant proportion of sub-optimal trajectories.  Specifically, we note below the percent of trajectories in each of the considered datasets that attain less than 50% of maximum observed return in each dataset:
> > > > > - Halfcheetah: 92.69%
> > > > > - AntWalker: 22.47%
> > > > > - Hill-climbing: 23.91%
> > > > > - Coordination game: 11.83%
> > > > >
> > > > > Note particularly that in the high-dimensional MultiAgentMuJoCo environments especially, a large proportion of low-reward trajectories exist in the dataset, with the HalfCheetah dataset being a particularly notable one where 92.69% of datasets attain less than 50% of maximum observed return.
> > > > >
> > > > > We additionally provide a more detailed overview of trajectory returns in each of Figs. 13a, 14a, 15a, and 16a.  These figures visualize the distribution of individual agent returns in each of the considered environments (respectively, hill-climbing, coordination game, 2-agent HalfCheetah, and 4-agent AntWalker). Notably, many of the trajectories in the HalfCheetah domain are skewed towards medium and low-return behaviors, with very few high-return / expert trajectories present (Fig. 13a). Interestingly, in the AntWalker dataset we observe a bimodal return distribution (seemingly consisting of low return and higher return trajectories, as seen in Fig. 14a). Closer inspection of this domain reveals that a large number of training runs cause highly suboptimal behaviors with very low returns; in Fig. 14b, we visualize the distribution of returns with these low-return trajectories excluded, observing a similar distribution to the HalfCheetah dataset (consisting primarily of medium-return trajectories). Similarly, a large proportion of trajectories with extremely low returns exist in the hill-climbing and coordination game domains (Figs. 15a and 16a); we verified that these low-return trajectories consist primarily of random agent behaviors collected early in training.
> > > > >
> > > > > As part of this investigation, we also analyze the discovered behavior spaces (both at joint and local agent levels) with respect to the return distributions. Specifically, Figs. 13b, 14c, 15b and 16b visualize the behavior spaces learned by MOHBA, which are then labeled by the ground truth return corresponding to each trajectory.  Interestingly, despite MOHBA not using any reward or return information in learning agent behavior spaces, clear clusters corresponding to low-return (random/early-training trajectories), medium-return trajectories, and expert return trajectories are automatically discovered in all of the considered domains; moreover, for trajectories with similar returns to one another, there is also diversity of the clusters they are categorized into (i.e., there exist multiple clusters with similar returns, and the trajectories are not simply being clustered by some proxy for return).  For example, in the hill-climbing environment, many of the trajectories corresponding to low returns are clustered in the center of the joint latent space in Fig. 15b; in the HalfCheetah environment, a prominent cluster of high-return trajectories is visible in the top-left and bottom-left local latent spaces of agents 0 and 1, respectively (Fig. 14c); in the AntWalker environment, the lowest-return trajectories are clustered in distinctive regions both at the joint and local level, with a medium-return trajectory set also visible in the latent spaces (Fig. 14c).

---

> > > > > > ### Author Response · Authors · 2022-08-02
> > > > > > **Author Response to Reviewer do8M (Part 6)**
> > > > > >
> > > > > > Overall, the above reward distribution analysis is helpful in determining that the analyzed datasets exhibit a wide range of trajectories (in terms of agent returns), covering random behaviors, sub-optimal behaviors, and high-return behaviors, similar to typical offline RL datasets. We thank the reviewer again for raising this concern. We have included the above discussion in Supplementary Information Section A.5 for reader benefit.
> > > > > >
> > > > > > —-------------------------—-------------------------—-----------
> > > > > >
> > > > > > 4) Minor issues: missing line of research on RL diversity.
> > > > > >
> > > > > > Response: We thank the reviewer for suggesting these relevant works. We now include this line of work in Section 2 (Related Works), and also include several additional papers from the RL diversity field:
> > > > > > - “Learning Self-Imitating Diverse Policies” (Gangwani et al., 2018)
> > > > > > - “Diversity-driven exploration strategy for deep reinforcement learning” (Hong et al., 2018)
> > > > > > - “Diverse exploration via conjugate policies for policy gradient methods” (Cohen at al., 2019)
> > > > > > - “Diversity-driven extensible hierarchical reinforcement learning” (Song et al., 2019)
> > > > > > - “Discovering diverse multi-agent strategic behavior via reward randomization” (Tang et al., 2021)
> > > > > >
> > > > > >  Note that despite their focus on policy diversity, several of these approaches make much stronger assumptions than ours (e.g., access to the individual policies for computation of the Shannon entropy term in Trajectory Diversity, and access to the policy-wise behavior embeddings for computing the similarity/kernel matrix in DvD). By contrast, our work assumes no access at all to any of the underlying raw policies, or even labels of which policies agent trajectories were obtained from. We agree that exploration of these approaches and their extensions to our considered regime would make for interesting future work, and hope our paper helps to further emphasize the importance of this area of study in the MARL community.
> > > > > >
> > > > > > —-------------------------—-------------------------—-----------
> > > > > >
> > > > > > 5) Minor issues: Experiments on ``user-desired characteristics'', conflation of “user-desired” with “high rewards”, and generally confusing nature of this part.
> > > > > >
> > > > > > Response: Thanks for providing this feedback. We apologize for the lack of clarity, and believe there are some potential misunderstandings here stemming from poor phrasing of this section on our part (namely, using the phrase “user-desired characteristics”, and not providing sufficient surrounding context of the section’s motivation). To clarify, the intent of this section is not to associate “user-desired” traits with “high reward” traits (which we agree should not to be conflated). Rather, its purpose is to:
> > > > > >
> > > > > > a) test the representation power of the latent spaces learned by MOHBA, by using the latent vectors z to make predictions of certain characteristics associated with agents. E.g., whether MOHBA’s latent z clusters trajectories with high and low reward despite not observing reward directly; or whether these latent vectors encode the degree of agent trajectory diversity as measured by ‘dispersion. We completely agree with the reviewer that supervised learning evaluation cannot be directly generalized to RL (and that is indeed not the focus of this section). Rather, we use the testing accuracy as a proxy for gauging the representation power of latent vectors z using a few characteristics of potential interest (reward and dispersion).
> > > > > >
> > > > > > b) illustrate a means of discovering ‘behavior concepts’ associated with the above characteristics directly in the latent space z itself, leveraging the technique of [13] as an illustrative means of doing so. This is where we used the term ‘user-desired characteristics’ (e.g., “what if a user sought behavior concepts that were associated with medium, rather than high, agent return”?), but agree this wording is confusing.
> > > > > >
> > > > > > Overall, we agree that the context of this section could have been much clearer in hindsight, and thank the reviewer for their constructive feedback. We have updated the wording of this section (and removed the mention of ‘user-desired characteristics’ altogether) to hopefully make this clearer.
> > > > > >
> > > > > > —-------------------------—-------------------------—-----------
> > > > > >
> > > > > > We thank the reviewer again for the helpful and constructive feedback. We hope our clarifications above, additional experiments, and revised PDF are helpful in addressing your concerns. We would be happy to discuss or clarify any further questions.

---

> > ### Comment · Reviewer_do8M · 2022-08-09
> > **I like the updated draft**
> >
> > I'm totally surprised by the updated draft and sincerely appreciate the efforts from the authors. This paper has reached its potential. :)
> >
> > I'm even looking forward to having an in-person conversation with the authors at NeurIPS.  Good job.

---

### Official Review · Reviewer_zatT · 2022-07-09

**Rating:** 7
**Confidence:** 4
**Soundness:** 3 good
**Presentation:** 3 good
**Contribution:** 3 good

**Summary:**

This paper introduces a method for discovery of behavior clusters (both individual-agent-level and joint-agent-level) in multiagent domains with the given offline trajectory, without assumption about the underlying learning algorithms and world models. Specifically, the method uses joint-level latent behavior factor z and individual-level latent behavior factor zi with the assumption that each agent’s latent-conditioned policy is conditionally-independent z given zi, and drive a variational lower bound to learn z and zi by maximizing the probabilities of the given offline trajectory probabilities. The authors illustrate the effectiveness of their method for enabling the coupled understanding of behaviors at the joint and local agent level, detection of behavior changepoints throughout training, discovery of core behavioral concepts (e.g., those that facilitate higher returns), and demonstrate the approach's scalability to a high-dimensional multiagent MuJoCo control domain.


**Questions:**

To be honest, I am not an expert of multiagent behavior analysis. I try to give some suggestions as follows.

* Could the authors give some reasons for the assumption that "each agent’s latent-conditioned policy is conditionally-independent z given zi"?
* I think it will be better to give more clear description about how to draw the figures of z and zi.
* I found that NCC-MARL [1] also proposes latent behavior analysis, and it shows that the consistent latent cognition of individual agents will result in good joint cooperation of all agents. I think this is also a representive paper that should be cited.


[1] Neighborhood cognition consistent multi-agent reinforcement learning. AAAI 2020.

**Limitations:**

-

**Strengths And Weaknesses:**

Strengths
* Good Research Point. The agent behavior analysis is less studied compared to the agent performance enhancement as far as I konw. I think the proposed method (which is less reward-centric) is very necessary for the multi-agent community.
* Clear Method and Good Paper Writting. The latent-conditioned trajectory generation process that the authors use to learn multiagent behavior clusters is very clear and easy to understand.
* Sufficient Experiments. Including both coupled analysis and indepedent analysis of both joint and local behavioral clusters analysis, emergent behavior analysis and the analysis of scalability to high-dimensional domains. And I also found that the figures is clear to understand.

Weaknesses
* I do not know whether the assumption that "each agent’s latent-conditioned policy is conditionally-independent z given zi" is true for real world scenarios? I understand the authors try to simplify the problem formulation, but I have some concerns for this assumption. For example (the same example used in the paper), if z encodes (at a high level) whether agents were cooperating or competing, I think the agent’s policy should certainly be dependent on z, no matter what meanings zi may represent.
* Since the paper focus on behavior analysis (ie, the analysis of z and zi), I think the authors should give clear description about how to draw the figures of z and zi, but some figures do not have such descriptions.

---

> ### Author Response · Authors · 2022-08-02
> **Author Response to Reviewer zatT (Part 1)**
>
> We thank the reviewer for the positive feedback regarding the importance of the behavior analysis regime to the multiagent community, the clarity of the paper, and the kind notes regarding the experiments.
>
> We have substantially improved the paper as a result of your feedback, and updated the paper PDF (which now includes both the main paper + Supplementary Information) accordingly, with changes summarized as follows:
> - Added new experiments for 4-agent OpenAI hide and seek policy checkpoints, where our method learns to distinguish ‘running and chasing’, ‘fort building’, ‘ramp use’, and ‘ramp defense’ behaviors [Supplementary Information A.3.1]
> - Added new experiments for 4-agent MultiAgent MuJoCo AntWalker [Supplementary Information A.3.1]
> - Conducted new experiments analyzing effects of key hyperparameter on results [Supplementary Information A.4]
> - Investigated the proportions of random, sub-optimal, and expert trajectories in our datasets, and conducted additional reward-based experiments, which revealed that MOHBA learns to cluster trajectories with similar agent returns despite not having access to reward information during training [Supplementary Information A.5]
> - Adding all suggested related works to discussions
> - Added numerous clarifications and added contexts throughout the text.
>
> Please note that due to the page limit still applying in the rebuttal period and due to limited time, we have included expanded experiments in the Supplementary Information, but intend to move them to the main text and move/shorten previously-existing sections to prioritize these latest results (especially on OpenAI hide and seek).
>
> Please find our point-by-point responses to your inquiries below:
>
> —-------------------------—-------------------------—-------------------------
>
> 1) “Assumption of latent-conditioned policy being conditionally-independent of z given zi” and “Could the authors give some reasons for the assumption that ‘each agent’s latent-conditioned policy is conditionally-independent z given zi’"?
>
> Response: We thank the reviewer for requesting this important clarification. (We assume that the reviewer is referring to the conditional independence of the policy $\pi^i$ from $z_{\omega}$, given $z_{\alpha}^i$.
>
> Please note that one of the prototypical paradigms in multiagent (MARL) training is that of ‘centralized training, decentralized execution’. In this standard regime, each agent’s decision-making policy is conditioned only on its available local information / local observations during execution; thus, the behaviors exhibited by each agent are ultimately informed by their local information, rather than global (joint) information. In our setting, $z_{\alpha}^i$ serves to provide this local behavior context for each agent. Thus, the assumption of the policy being conditionally-independent of $z_{\omega}$ given $z_{\alpha}^i$ corresponds well to the assumption of agents only using local information (rather than joint information) in MARL to inform their policy/decision-making.
>
> Nonetheless, we also note that there is a strong relationship between local and joint behaviors in our setting, which provides a coordination signal to the policy implicitly through $z_{\alpha}^i$. Specifically, our derivation results in a local behavior prior term, $p(z_{\alpha}^i | z_{\omega})$; using this term, the joint behavior latent $z_{\omega}$ is able to influence the learned space over local behavior latents $z_{\alpha}^i$. Thus, the information-flow in our framework can be more intuitively described as follows:
>
> a) The joint behavior latent observes a trajectory and summarizes the joint behavior exhibited in it (e.g., team-wide cooperation, competition, etc.) in $z_{\omega}$
> b) Subsequently, this joint latent $z_{\omega}$ affects the space of $z_{\alpha}^i$, which are then sampled and used to inform each agent how to behave locally in order to achieve the joint behavior observed.
>
> Overall, the above relationship between the two latent spaces is why we believe the described conditional-independence of the policy is a fairly reasonable assumption. We have now included this discussion in Supplementary Information Section A.1.1 for further clarity.

---

> > ### Author Response · Authors · 2022-08-02
> > **Author Response to Reviewer zatT (Part 2)**
> >
> > —-------------------------—-------------------------—-------------------------
> >
> > 2) Need for clearer description about how to draw the figures of z and zi.
> >
> > Response: Thank you for suggesting this, we completely agree that given the important role these figures play in the paper, a clearer description will improve the paper’s clarity. We now include this in Supplementary Information A.6, which we also paste below for convenience:
> >
> > “The visualization procedure we use is as follows:
> > 1.  We first train MOHBA using the offline behavior datasets of interest.
> > 2. Following training of MOHBA, we pass each trajectory $\tau$ through both the joint encoder $q_{\phi}(z_{\omega}|\tau)$ and the agent-wise local encoders $q_{\phi}(z_{\alpha}^i|\tau)$, sampling a joint latent $z_{\omega}$ and a set of local latents $z_{\alpha}^i$ for all $N$ agents accordingly. Thus, for $K$ such trajectories we obtain a set of latent parameters $[{(z_{\omega,k}, z_{\alpha,k}^1,\cdots,z_{\alpha,k}^N)}]_{k=1}^{K}$.
> > 3. As the latent parameters are high-dimensional in nature, we visualize their 2D projection (e.g., using Principal Component Analysis), thus yielding the behavior space visualization such as those in Fig. 3a.”
> >
> > —-------------------------—-------------------------—-------------------------
> >
> > 3) Citing NCC-MARL (Neighborhood cognition consistent multi-agent reinforcement learning. AAAI 2020)
> >
> > Response: We thank the reviewer for suggesting this relevant paper. We now discuss and cite it in our related works section.
> >
> > —-------------------------—-------------------------—-------------------------
> >
> > We thank the reviewer again for the helpful and constructive feedback. We hope our clarifications above, additional experiments, and revised PDF are helpful in addressing your concerns. We would be happy to discuss or clarify any further questions.

---

> > > ### Comment · Reviewer_zatT · 2022-08-09
> > > **Thanks for the responses.**
> > >
> > > I thank the authors for the detailed responses. They addressed all my concerns.

---

### Official Review · Reviewer_c5wJ · 2022-07-11

**Rating:** 6
**Confidence:** 3
**Soundness:** 3 good
**Presentation:** 3 good
**Contribution:** 3 good

**Summary:**

The paper focuses on behavior analysis of multiagent reinforcement learning. The approach is to discover behaviors at the local level and joint level separately from offline data based on an information-theoretical objective. The authors did an empirical analysis on two simple domains as well as the multiagent mujoco domain. The results show some interesting multiagent behaviors visualized in the learned latent space.

**Questions:**

For the 3-agent hill climbing environment, it seems to me that the goal of the domain is more like to train the three agents independently (reach one of the three goal positions) and there's no multiagent cooperation/interactions happening.

I don't understand the LSTM baseline on page 7. Is it more like an ablation study of the network structure?

It would be nice to see some ablation study on the KL weighting term $\beta$ in the algorithm's main training objective. Besides, how will the size of the latent vector $z$ affect the overall results?

Another suggestion (probably for future work) is that maybe the authors can try to let the agent learn some transferable multiagent "skills" using the proposed method and see if they can be reused to solve some other similar tasks.


**Limitations:**

See above.

**Strengths And Weaknesses:**

The high-level idea of analyzing multiagent behaviors is appealing and I think is an important direction for multiagent RL research. The authors propose to learn hierarchical latent embeddings of the agents' behaviors by focusing on the local and joint level separately. The proposed approach is simple and easy to follow. The empirical analysis is thorough by first analyzing the two levels of clusters independently and then their relationships.

I think one of the weaknesses of the submission is lack of empirical analysis on more complex domains that requires structured cooperation/competition strategy, like simulated football game. It would be interesting to see how different categories of skills (e.g. dribbling the ball) look like in latent space. I'm also a little doubtful whether the information-theoretical training objective will work properly for such domains. Besides, the first domain of empirical analysis does not seem to be a typical multiagent environment with interactions between agents.

---

> ### Author Response · Authors · 2022-08-02
> **Author Response to Reviewer c5wJ (Part 1)**
>
> We thank the reviewer for the positive feedback regarding the multiagent behavior analysis regime being an important area of study, and the clarity of our proposed approach. We also appreciate your constructive feedback, especially regarding the need for further evaluation, which we have focused our efforts on during the rebuttal period.
>
> We have substantially improved the paper as a result of your feedback, and updated the paper PDF (which now includes both the main paper + Supplementary Information) accordingly, with changes summarized as follows:
> - Added new experiments for 4-agent OpenAI hide and seek policy checkpoints, where our method learns to distinguish ‘running and chasing’, ‘fort building’, ‘ramp use’, and ‘ramp defense’ behaviors [Supplementary Information A.3.1]
> - Added new experiments for 4-agent MultiAgent MuJoCo AntWalker [Supplementary Information A.3.1]
> - Conducted new experiments analyzing effects of key hyperparameter on results [Supplementary Information A.4]
> - Investigated the proportions of random, sub-optimal, and expert trajectories in our datasets, and conducted additional reward-based experiments [Supplementary Information A.5]
> - Adding all suggested related works to discussions
> - Added numerous clarifications and added contexts throughout the text.
>
> Please note that due to the page limit still applying in the rebuttal period and due to limited time, we have included these expanded experiments in the supplementary information of the above PDF, but intend to move them to the main text and move/shorten previously-existing sections to prioritize these latest results. All figure & table # references below refer to this updated revision. Please find our point-by-point responses below for details.

---

> > ### Author Response · Authors · 2022-08-02
> > **Author Response to Reviewer c5wJ (Part 2)**
> >
> > 1) Need for additional evaluations.
> >
> > Response: We agree that the hill-climbing environment is indeed simplistic, we primarily intended it to be used as a motivating domain (i.e., a useful sanity check) with easily-explainable that could be used to validate behavioral results in a controlled setting. As a result of your and other reviewers’ feedback, during the rebuttal period, we substantially expanded the experiments section, including the addition of:
> >
> > a) results for two large-scale new domains, including one with policies trained by external authors (OpenAI hide and seek, visualized by the original authors here: https://www.youtube.com/watch?v=kopoLzvh5jY) and an expansion of the Multiagent MuJoCo results in the larger 4-agent AntWalker domain [revised Supplementary Information A.3.1]
> >
> > b) further ablative analysis of our existing datasets (including effects of hyperparameters on learned latent spaces) [revised Supplementary Information A.4]
> >
> > c) analysis of our previous latent spaces from a return-centric perspective, which reveals that MOHBA (our algorithm) learns to cluster trajectories with similar agent returns despite not having access to reward information during training [revised Supplementary Information A.5]
> >
> > In detail:
> >
> > a) First, per your suggestion, we analyzed agents interacting in more complex domains. Specifically, during the rebuttal period we were able to access and analyze policy checkpoints from the OpenAI hide-and-seek environment (https://github.com/openai/multi-agent-emergence-environments). We summarize results in Section A.3.1 of the Supplementary Information, which we also include below for convenience:
> >
> > """
> >
> > In the OpenAI hide-and-seek environment, two teams of two agents (hiders and seekers) compete against one another in a rich, high-dimensional environment with various dynamic objects (boxes and ramps) that the agents can interact with. OpenAI has open-sourced policy check-points manually annotated by human experts as exhibiting various distinctive multiagent behaviors at key stages of training in github.com/openai/multi-agent-emergence-environments.  We consider four distinctive policy checkpoints that each correspond to the following human-annotated labels:
> >
> > A) ‘running and chasing’, B) ‘fort building’, C) ‘ramp use’, and D) ‘ramp defense’.
> >
> > The state-space for each of the agents at each timestep of the trajectory is 100-dimensional, consisting of its own state (position, rotation, and velocity information), states of the other 3 agents, and the states of 3 boxes and one ramp in the environment (position, velocity, and box size information); each agent’s action space consists of a 3-dimensional force vector, and a ‘glue’ and ‘lock’ action for interacting with other objects. Overall, this yields a state-action space with dimensionality (4 agents x 105 state-actions), significantly larger than the (2 agents x 9 state-actions) of the HalfCheetah MuJoCo environment. We collect 100 trajectories per policy checkpoint, with each trajectory being 200 decision-steps long. Note that these trajectories have a wide distribution of behaviors, as agent and object initializations are also random in each episode.
> >
> > We then mix the trajectories collected from all of the above policy checkpoints together, then train MOHBA on this shuffled dataset. As usual, note that we provide no reward or return information to MOHBA. Figure 7 visualizes the behavior spaces discovered by MOHBA in the hide-and-seek domain. Agents 0 and 1 in this figure correspond to ‘hiders’, whereas agents 2 and 3 are ‘seekers’. We label each of the trajectories in this figure with the human-annotated behavior labels provided by OpenAI.
> >
> > In Fig. 7, we observe the presence of clear behavior clusters that correspond well to the expert policy checkpoint labels, both at the joint and local agent levels. Note that this is despite all the policy checkpoint data being shuffled in the dataset being used to train MOHBA.
> >
> > Interestingly, for the seekers (agents 2 and3), policy A (‘running and chasing’) is highly distinctive and well-separated from the other behaviors. Interestingly, despite the order of emergent behaviors in the original hide-and-seek MARL training being A→B→C→D, policies B (‘fort building’) and D (‘ramp defense’) appear to be behaviorally slightly closer to one another than C (‘ramp use’) and D, namely in the joint space and also in that of the seeker (agent 2 and 3) latent spaces. One hypothesis is that this could perhaps be due to both the ‘fort building’ and ‘ramp defense’ policies being associated with situations where the seekers cannot easily find the hiders, due to the hiders using obstacles to block entrances (B) and moving ramps to prevent their effective use (D).
> >
> > Overall, these experiments help to validate MOHBA’s learned latent spaces using policy labels manually annotated by human experts, and also highlight the applicability of our algorithm in domains with large state spaces.
> > """

---

> > > ### Author Response · Authors · 2022-08-02
> > > **Author Response to Reviewer c5wJ (Part 3)**
> > >
> > > We additionally expanded our existing MultiAgent MuJoCo experiments to the AntWalker domain (also summarized in Section A.3.1 of the Supplementary Information). Here, 4 agents each control one of 4 ant legs to coordinate movement towards the +x-direction (Fig. 6a).  To collect data for this domain, we conduct a wider MARL parameter sweep using both the TD3 and SAC algorithms, with varying training batch sizes and learning rates to gather a widely-varying dataset of behaviors. We subsequently train MOHBA on this data, visualizing the learned behavior clusters in Fig. 6b. We observe several behavior clusters of interest; notably, a large joint cluster exists in the top-right region of the joint and local returns, which, upon inspection of underlying trajectory videos, corresponds to cases where agents attain extremely low return. Similarly, there exists a smaller cluster in the lower-left region of the joint returns that also attains extremely low performance. The remaining clusters correspond to the agents displaying various ant-poses, andmoving only incrementally. One exceptional cluster also exists in the left region of the joint behavior space, which attains medium-level return (points that are primarily red in color, seen in the bottom left region of Figure 14’s joint return plots). On closer inspection, the AntWalker behavior in this cluster corresponds to one of the agents learning a reasonably good walking gait, while the remaining three agents remain stationary.
> > >
> > > b) Next, we also now conduct ablative analysis to understand the effects of key hyperparameters (the latent dimensionality size, and the KL loss $\beta$ term) on the learned latent spaces, and include this in Supplementary Materials Section A.4. Specifically, we compare both the policy reconstruction loss (Eq. 5) across ablations to attain a quantitative comparison of the quality of the latent encodings, and additionally compare the structure of the learned latent spaces themselves as a function of these parameters for a qualitative understanding. At a high level, we find that the sensitivity of the results to these hyperparameters is fairly low (in the range of values explored in these experiments), thus allowing the high level behaviors discovered to remain reasonably distinctive across the various sweeps, as seen in Figures 10 to 12. Please refer to Supplementary A.4 for figures and significant additional details.
> > >
> > > c) We additionally investigate the structure of the latent space from a reward-centric perspective, which reveals several novel insights that we summarize in Supplementary Information Section A.5. During the rebuttal period, we conducted additional analysis of MOHBA’s latent spaces, and discovered that MOHBA learns clear clusters corresponding to low-return (random/early-training) trajectories, medium-return trajectories, and expert return trajectories across all of the domains we considered; this is despite MOHBA never observing any agent reward information during training.
> > >
> > > These results help indicate the richness of the learned latent spaces in terms of not only raw trajectories but also their capacity to cluster trajectories that exhibit similar performance, even without observing the reward function.
> > >
> > > Overall, we believe these substantially-expanded results help to further validate our general methodology, quality of the datasets analyzed, and applicability to wider settings including policies collected from external open-sourced codebases.
> > >
> > > —-------------------------—-------------------------—-----------
> > >
> > > 2) I don't understand the LSTM baseline on page 7. Is it more like an ablation study of the network structure?
> > >
> > > Response: Thanks for this clarifying question. The LSTM model is meant to be a standalone baseline that targets using the LSTM hidden states, rather than a learned distribution over latent variables z, for understanding and clustering agent behaviors. This is a strong standalone baseline that is akin to typical techniques used in previous MARL studies, wherein LSTM-based agents are first trained to conduct a particular task, and their hidden states are subsequently analyzed to discover clusters of certain behaviors or representational power (in terms of linearly predicting certain characteristics of the environment). Examples of such analysis approaches include “Human-level performance in first-person multiplayer games with population-based deep reinforcement learning” (Jaderberg et al., 2018; Figure 3) and “From Motor Control to Team Play in Simulated Humanoid Football” (Liu et al., 2022; Figure 6b). We have updated the baselines section of the Experiments to provide this surrounding context.

---

> > > > ### Author Response · Authors · 2022-08-02
> > > > **Author Response to Reviewer c5wJ (Part 4)**
> > > >
> > > > —-------------------------—-------------------------—-----------
> > > >
> > > > 3) It would be nice to see some ablation study on the KL weighting term β in the algorithm's main training objective. Besides, how will the size of the latent vector z affect the overall results?
> > > >
> > > > Response: We thank the reviewer for suggesting this. We conducted this ablative analysis per your suggestion and include it in Supplementary Materials Section A.4. We include these findings below for convenience:
> > > >
> > > > “We focus our analysis on the KL-loss $\beta$ term in the training objective, alongside the latent $z_{\omega}$ and $z_{\alpha}$ dimensionality. We both compare the policy reconstruction loss (Eq. 5) across ablations to attain a quantitative comparison of the quality of the latent encodings, and additionally compare the structure of the learned latent spaces themselves as a function of these parameters for a qualitative understanding.
> > > >
> > > > First, in terms of the raw policy reconstruction loss, we find in Figs. 10a, 11a and 12a that the latent dimensionality has a higher impact on policy reconstruction performance than the weighing term $\beta$; intuitively, increasing latent dimensionality leads to better reconstruction (lower loss) due to the latent space being able to encode more behavioral information about the agent trajectories. By contrast, the effects of the KL weighing term are more negligible in terms of reconstruction loss.
> > > >
> > > > Second, we inspect the effects of these hyperparameter sweeps on the learned latent distributions themselves. Specifically, the respective panels (b)-(e) of each of Fig. 10,  Fig. 11,  and Fig. 12 visualize the change in latent space structure as a function of the latent dimensionality and $\beta$. At a high-level, prominent behavioral clusters mentioned in the main text are re-discovered throughout these parameter sweeps (e.g., the three local clusters for the hill climbing environment, the joint and local clusters for the coordination environment, and the various walking gait clusters for the HalfCheetah environment). For the more complex latent spaces (e.g., HalfCheetah), increasing the dimensionality of z tends to increase the number of joint clusters identified (e.g., compare Fig. 12c versus Fig. 12e); this is intuitive as a larger latent space results in a richer encoding, capable of distinguishing more nuanced behaviors. Increasing the KL $\beta$ term from 1e−4 to 1e−2 tends to result in clusters that overlap more / are ‘softer’ and slightly less distinguishable (e.g., comparingzωin Fig. 12c versus Fig. 12d); this also aligns well with intuition, as increasing the $\beta$ term prioritizes the KL divergence between the posterior and prior, which deprioritizes disentangling the behaviors for the policy reconstruction term (5).
> > > >
> > > > Overall, these results provide us intuition in terms of the role of these hyperparameters in the behavior clusters learned. At a high level, it appears that the sensitivity of the results to these hyperparameters is fairly low (in the range of values explored in these experiments), thus allowing the high level behaviors discovered to remain reasonably distinctive across the various sweeps, as seen in Figures 10 to 12.”

---

> > > > > ### Author Response · Authors · 2022-08-02
> > > > > **Author Response to Reviewer c5wJ (Part 5)**
> > > > >
> > > > > —-------------------------—-------------------------—-----------
> > > > >
> > > > > 4) Another suggestion (probably for future work) is that maybe the authors can try to let the agent learn some transferable multiagent "skills" using the proposed method and see if they can be reused to solve some other similar tasks.
> > > > >
> > > > > Response: Thanks for the suggestion. We agree, this is a great idea and a direct benefit of the generative nature of our framework. We did not include this in the original submission as we wanted to place focus on the behavior analysis domain itself as this is a novel problem regime; we thus first wanted to validate this direction of research and thus encourage the community to prioritize this under-studied domain. As suggested by the reviewer, the MOHBA framework is indeed amenable to transfer of multiagent behaviors. To expand a bit, as we learn a distribution over both the joint and local behavior latent parameters, one may achieve this type of downstream transfer by either:
> > > > >
> > > > > a) learning a meta-policy that samples joint/coordinated behaviors in $z_{\omega}$ (joint latent) space, then passes them to the local skill encoder, and finally to MOHBA’s latent-conditioned policies $\pi(a^i|s, z_{\alpha})$.
> > > > >
> > > > > b) learning independent meta-policies for each agent, which would directly output $z_{\alpha}$ for each agent, and subsequently pass them to the latent-conditioned policies.
> > > > >
> > > > > In both cases, this outer meta-policy could then be trained via standard RL algorithms.
> > > > >
> > > > > One related question here is whether MOHBA learns to disentangle trajectories that yield high and low rewards (which would be potentially useful for this downstream transfer learning), despite never observing any reward information in trajectories. During the rebuttal period, we conducted additional analysis of MOHBA’s latent spaces, and discovered that MOHBA indeed learns clear clusters corresponding to low-return (random/early-training) trajectories, medium-return trajectories, and expert return trajectories across all of the domains we considered. This might serve as a useful characteristic for future follow ups on transfer learning, where the meta-policy can potentially learn to exploit high-reward regions of the latent space for sample-efficient transfer.
> > > > >
> > > > > Thanks again for suggesting this interesting line of thought. We now include this in the future works paragraph of the Discussion section of the revision.
> > > > >
> > > > > —-------------------------—-------------------------—-----------
> > > > >
> > > > > We thank the reviewer again for the helpful and constructive feedback. We hope our clarifications above, additional experiments, and revised PDF are helpful in addressing your concerns. We would be happy to discuss or clarify any further questions.

---

### Official Review · Reviewer_Nrc6 · 2022-07-12

**Rating:** 7
**Confidence:** 5
**Soundness:** 3 good
**Presentation:** 4 excellent
**Contribution:** 3 good

**Summary:**

The authors develop a hierarchical clustering technique for sequential data and apply it to RL agents by optimizing a variational objective. The method is tested on simple multi-agent environments including a MuJoCo example.


**Questions:**

What is the sample complexity of this approach? How many traces of behavior are needed to get good behavioral clusters?

Is access and control of the environment required or does one just need traces of behavior?

How will this algorithm perform in environments that have stochastic transitions?

How does this model differ from a generic model for clustering sequences? Is there something in the model structure that makes it more likely to detect behavioral signatures as opposed to physical signatures?


**Limitations:**

Yes

**Strengths And Weaknesses:**

The paper is very well written and easy to follow. The methods are explained clearly and each experiment and example is well motivated. All results are discussed and the figures are clear and well explained.

The problem is an important and impactful one and progress here would be exciting to many in the MARL community.

However, the current work suffers from the following weaknesses:

Too much time spent on the hill-climbing problem. While this problem is very intuitive, it has almost no distinctly multiagent content (interaction between the agents) and thus makes it hard to see how this approach will handle more interesting kinds of multi-agent behavior. For example, the authors allude to "cooperate vs. compete" but there is no analysis of sophisticated behavior.

The evaluation of the method needs significantly more work. To raise my score, I would like to see is the analysis of an existing agent (not trained by the authors) that demonstrates some kind of sophisticated behavior and to have the model presented here discover some kind of novel insight into what is going on. It would be particularly interesting for example to analyze a complete system with its ablations to give finer-grained insight into why an ablation doesn't work as well.

UPDATE: Scores raised in response to new experiments and the answers to my questions.

---

> ### Author Response · Authors · 2022-08-02
> **Author Response to Reviewer Nrc6 (Part 1)**
>
> We thank the reviewer for the positive feedback regarding the paper clarity and the importance of the problem setting studied.  We also appreciate your constructive feedback, especially regarding the need for further evaluation, which we have focused our efforts on during the rebuttal period.
>
> We have substantially improved the paper as a result of your feedback, and updated the paper PDF (which now includes both the main paper + Supplementary Information) accordingly, with changes summarized as follows:
> - Added new experiments for 4-agent OpenAI hide and seek policy checkpoints, where our method learns to distinguish ‘running and chasing’, ‘fort building’, ‘ramp use’, and ‘ramp defense’ behaviors [Supplementary Information A.3.1]
> - Added new experiments for 4-agent MultiAgent MuJoCo AntWalker [Supplementary Information A.3.1]
> - Conducted new experiments analyzing effects of key hyperparameter on results [Supplementary Information A.4]
> - Investigated the proportions of random, sub-optimal, and expert trajectories in our datasets, and conducted additional reward-based experiments [Supplementary Information A.5]
> - Adding all suggested related works to discussions
> - Added numerous clarifications and added contexts throughout the text.
>
>  Please note that due to the page limit still applying in the rebuttal period and due to limited time, we have included expanded experiments in the Supplementary Information, but intend to move them to the main text and move/shorten previously-existing sections to prioritize these latest results (especially on OpenAI hide and seek). All figure & table # references below refer to this updated revision.
>
> Please find our point-by-point responses below for details.

---

> > ### Author Response · Authors · 2022-08-02
> > **Author Response to Reviewer Nrc6 (Part 2)**
> >
> > 1) Need for additional evaluations.
> >
> > Response: We agree that the hill-climbing environment is indeed simplistic, we primarily intended it to be used as a motivating domain (i.e., a useful sanity check) with easily-explainable that could be used to validate behavioral results in a controlled setting. As a result of your and other reviewers’ feedback, during the rebuttal period, we substantially expanded the experiments section, including the addition of:
> >
> > a) results for two large-scale new domains, including one with policies trained by external authors (OpenAI hide and seek, visualized by the original authors here: https://www.youtube.com/watch?v=kopoLzvh5jY) and an expansion of the Multiagent MuJoCo results in the larger 4-agent AntWalker domain [revised Supplementary Information A.3.1]
> >
> > b) further ablative analysis of our existing datasets (including effects of hyperparameters on learned latent spaces) [revised Supplementary Information A.4]
> >
> > c) analysis of our previous latent spaces from a return-centric perspective, which reveals that MOHBA (our algorithm) learns to cluster trajectories with similar agent returns despite not having access to reward information during training [revised Supplementary Information A.5]
> >
> > In more detail:
> >
> > a) First, per your suggestion, we analyzed existing agents not trained by us. Specifically, during the rebuttal period we were able to access and analyze policy checkpoints from the OpenAI hide-and-seek environment (https://github.com/openai/multi-agent-emergence-environments). We summarize results in Section A.3.1 of the Supplementary Information, which we also include below for convenience:
> >
> > """
> > In the OpenAI hide-and-seek environment, two teams of two agents (hiders and seekers) compete against one another in a rich, high-dimensional environment with various dynamic objects (boxes and ramps) that the agents can interact with. OpenAI has open-sourced policy check-points manually annotated by human experts as exhibiting various distinctive multiagent behaviors at key stages of training in github.com/openai/multi-agent-emergence-environments.  We consider four distinctive policy checkpoints that each correspond to the following human-annotated labels:
> > A) ‘running and chasing’, B) ‘fort building’, C) ‘ramp use’, and D) ‘ramp defense’.
> >
> > The state-space for each of the agents at each timestep of the trajectory is 100-dimensional, consisting of its own state (position, rotation, and velocity information), states of the other 3 agents, and the states of 3 boxes and one ramp in the environment (position, velocity, and box size information); each agent’s action space consists of a 3-dimensional force vector, and a ‘glue’ and ‘lock’ action for interacting with other objects. Overall, this yields a state-action space with dimensionality (4 agents x 105 state-actions), significantly larger than the (2 agents x 9 state-actions) of the HalfCheetah MuJoCo environment.
> >
> > We collect 100 trajectories per policy checkpoint, with each trajectory being 200 decision-steps long. Note that these trajectories have a wide distribution of behaviors, as agent and object initializations are also random in each episode.
> >
> > We then mix the trajectories collected from all of the above policy checkpoints together, then train MOHBA on this shuffled dataset. As usual, note that we provide no reward or return information to MOHBA. Figure 7 visualizes the behavior spaces discovered by MOHBA in the hide-and-seek domain. Agents 0 and 1 in this figure correspond to ‘hiders’, whereas agents 2 and 3 are ‘seekers’. We label each of the trajectories in this figure with the human-annotated behavior labels provided by OpenAI.
> >
> > In Fig. 7, we observe the presence of clear behavior clusters that correspond well to the expert policy checkpoint labels, both at the joint and local agent levels. Note that this is despite all the policy checkpoint data being shuffled in the dataset being used to train MOHBA.
> >
> > Interestingly, for the seekers (agents 2 and3), policy A (‘running and chasing’) is highly distinctive and well-separated from the other behaviors. Interestingly, despite the order of emergent behaviors in the original hide-and-seek MARL training being A→B→C→D, policies B (‘fort building’) and D (‘ramp defense’) appear to be behaviorally slightly closer to one another than C (‘ramp use’) and D, namely in the joint space and also in that of the seeker (agent 2 and 3) latent spaces. One hypothesis is that this could perhaps be due to both the ‘fort building’ and ‘ramp defense’ policies being associated with situations where the seekers cannot easily find the hiders, due to the hiders using obstacles to block entrances (B) and moving ramps to prevent their effective use (D).
> > """
> >
> > Overall, these experiments help to validate MOHBA’s learned latent spaces using policy labels manually annotated by human experts, and also highlight the applicability of our algorithm in domains with large state spaces.

---

> > > ### Author Response · Authors · 2022-08-02
> > > **Author Response to Reviewer Nrc6 (Part 3)**
> > >
> > > We additionally expanded our existing MultiAgent MuJoCo experiments to the AntWalker domain (also summarized in Section A.3.1 of the Supplementary Information). Here, 4 agents each control one of 4 ant legs to coordinate movement towards the +x-direction (Fig. 6a).  To collect data for this domain, we conduct a wider MARL parameter sweep using both the TD3 and SAC algorithms, with varying training batch sizes and learning rates to gather a widely-varying dataset of behaviors. We subsequently train MOHBA on this data, visualizing the learned behavior clusters in Fig. 6b. We observe several behavior clusters of interest; notably, a large joint cluster exists in the top-right region of the joint and local returns, which, upon inspection of underlying trajectory videos, corresponds to cases where agents attain extremely low return. Similarly, there exists a smaller cluster in the lower-left region of the joint returns that also attains extremely low performance. The remaining clusters correspond to the agents displaying various ant-poses, and moving only incrementally. One exceptional cluster also exists in the left region of the joint behavior space, which attains medium-level return (points that are primarily red in color, seen in the bottom left region of Figure 14’s joint return plots). On closer inspection, the AntWalker behavior in this cluster corresponds to one of the agents learning a reasonably good walking gait, while the remaining three agents remain stationary.
> > >
> > > b) Next, we also now conduct ablative analysis to understand the effects of key hyperparameters (the latent dimensionality size, and the KL loss $\beta$ term) on the learned latent spaces, and include this in Supplementary Materials Section A.4. Specifically, we compare both the policy reconstruction loss (Eq. 5) across ablations to attain a quantitative comparison of the quality of the latent encodings, and additionally compare the structure of the learned latent spaces themselves as a function of these parameters for a qualitative understanding. At a high level, we find that the sensitivity of the results to these hyperparameters is fairly low (in the range of values explored in these experiments), thus allowing the high level behaviors discovered to remain reasonably distinctive across the various sweeps, as seen in Figures 10 to 12. Please refer to Supplementary A.4 for figures and significant additional details.
> > >
> > > c) We additionally investigate the structure of the latent space from a reward-centric perspective, which reveals several novel insights that we summarize in Supplementary Information Section A.5. During the rebuttal period, we conducted additional analysis of MOHBA’s latent spaces, and discovered that MOHBA learns clear clusters corresponding to low-return (random/early-training) trajectories, medium-return trajectories, and expert return trajectories across all of the domains we considered; this is despite MOHBA never observing any agent reward information during training. Overall, these results indicate the richness of the learned latent spaces in terms of not only raw trajectories but also their capacity to cluster trajectories that exhibit similar performance, even without observing the reward function.
> > >
> > > Overall, we believe these substantially-expanded results help to further validate our general methodology, quality of the datasets analyzed, and applicability to wider settings including policies collected from external open-sourced codebases.
> > >
> > > —-------------------------—-------------------------—-----------
> > >
> > > 2) What is the sample complexity of this approach? How many traces of behavior are needed to get good behavioral clusters?
> > >
> > > Response: Datasets collected and analyzed in our experiments were comparable in size to offline RL datasets, consisting of 25k trajectories for the hill-climbing domain, 20k trajectories for the coordination domain, and 100k trajectories for the MuJoCo HalfCheetah domain. However, during the rebuttal phase, we were able to run experiments in the OpenAI hide and seek domain, where were were able to disentangle agent behaviors using significantly smaller datasets consisting of 400 multiagent trajectories in total (4 policy checkpoints, 100 trajectories collected from each). Based on these latest results, we expect the approach to work well with (roughly speaking) on the order of 100s to 1000s of trajectories in total (with higher fidelity likely to be obtained with more trajectories).
> > >
> > > —-------------------------—-------------------------—-----------
> > >
> > > 3) Is access and control of the environment required or does one just need traces of behavior?
> > >
> > > Response: Our method requires no access or control of the environment at all. It simply uses the raw state-action trajectories / policy rollouts of agents collected offline (with no reward information needed), which is one of its primary benefits. Thanks for raising this point, we have updated the paper introduction to explicitly mention this.

---

> > > > ### Author Response · Authors · 2022-08-02
> > > > **Author Response to Reviewer Nrc6 (Part 4)**
> > > >
> > > > —-------------------------—-------------------------—-----------
> > > >
> > > > 4) How will this algorithm perform in environments that have stochastic transitions?
> > > >
> > > > Response: A core benefit of the variational inference backbone of our method is its ability to handle stochasticity in the dataset while disentangling agent behaviors, through a combination of two features:
> > > >
> > > > a) the policy reconstruction term in the algorithm, which emphasizes the use of latent vectors z to disentangle and learn agent actions, rather than state-transitions [further details provided in the response to the reviewer’s subsequent question];
> > > >
> > > > b) the VAE-backbone of the method, which should help MOHBA cope well in stochastic settings.
> > > >
> > > > We also note that even in the deterministic environments we considered, the datasets we analyze were collected from numerous MARL training runs in parallel, with trajectories collected throughout all of training; as such, all of our considered datasets are composed of a mixture of random behaviors (collected from the beginning phase of training), sub-optimal behaviors (at intermediate checkpoints in training), and near-optimal behaviors (from end of training), where agents can exhibit different behaviors in each trajectory even when in identical states. Moreover, several of the environments we consider are randomly initialized (e.g., agent and object positions are randomly initialized in the OpenAI hide and seek environment we analyze in the rebuttal phase), further adding to the stochasticity of the considered datasets.
> > > >
> > > > —-------------------------—-------------------------—-----------
> > > >
> > > > 5) How does this model differ from a generic model for clustering sequences? Is there something in the model structure that makes it more likely to detect behavioral signatures as opposed to physical signatures?
> > > >
> > > > Response: Great question, please note that we indeed do test fairly strong sequential clustering approaches, such as the LSTM baseline (see below for details). However, one of the primary differences between our model and traditional clustering approaches is the policy reconstruction component; this term influences the model to use the latent information (z_alpha^i for each agent i) to select actions that are as similar as possible to those originally exhibited by said agent in the trajectory dataset. As such, the approach is less focused on reconstruction of the full trajectories / state-transitions, and more so on the reconstruction of the agent behaviors themselves, as captured through actions. Additionally, as observed in the sequential clustering LSTM baseline (which has been previously used for behavior analysis in MARL works such as [1] and [2] below), the disentanglement of trajectory-encoding latents from the policy head (as done in MOHBA) seems to significantly outperform a pure sequential clustering approach (LSTM), as observed in Figures 8 and 9 of the paper.
> > > >
> > > > An important closely related note is that one of the potential benefits of this policy-reconstruction approach is that it enables the potential re-use of the learned latent-conditioned policy for downstream tasks. I.e., as the latent-conditioned policy is one of the artifacts resulting from training MOHBA, these policies could be used in related tasks where agents explore the latent behavior spaces, rather than raw action spaces, to transfer knowledge to new domains. Such an approach is similar to existing single-agent skill learning approaches, and although is not a focus of this particular iteration of our behavior analysis work, it may make for an interesting direction of future research.
> > > >
> > > > [1]  “Human-level performance in first-person multiplayer games with population-based deep reinforcement learning” (Jaderberg et al., 2018; Figure 3)
> > > >
> > > > [2] “From Motor Control to Team Play in Simulated Humanoid Football” (Liu et al., 2022; Figure 6b)
> > > >
> > > > —-------------------------—-------------------------—-----------
> > > >
> > > > We thank the reviewer again for the helpful and constructive feedback. We hope our clarifications above, additional experiments, and revised PDF are helpful in addressing your concerns. We would be happy to discuss or clarify any further questions.

---

> > > > > ### Comment · Reviewer_Nrc6 · 2022-08-07
> > > > > **Thank you for the response**
> > > > >
> > > > > Thank you for the detailed responses. My concerns have been fully addressed and I now support the publication of this manuscript. I have raised the scores in my review accordingly.

---

### Meta-Review · Area_Chair_1nm7 · 2022-08-21

**Recommendation:** Accept
**Confidence:** Certain

**Metareview:**

The reviewers agreed this paper studies an interesting problem and provide an interesting contribution to the multi-agent community. We urge the authors to include the added experiments and information (e.g., suggested related work) into the main text.

**Award:**

No

---

### Decision · Program_Chairs · 2022-09-14

Accept